# The Bern Simple Climate Model (BernSCM) v1.0: an extensible and fully documented open source reimplementation of the Bern reduced form model for global carbon cycle-climate simulations

Kuno Strassmann[1,2] and Fortunat Joos[1,3]

[1]Climate and Environmental Physics, Physics Institute, University of Bern, Bern, Switzerland
[2]now at Federal Institute of Technology, Switzerland
[3]Oeschger Center for Climate Change Research, University of Bern, Bern, Switzerland.

*Correspondence to:* Kuno Strassmann (kuno.strassmann@alumni.ethz.ch)

**Abstract.** The Bern Simple Climate Model (BernSCM) is a free open source reimplementation of a reduced form carbon cycle-climate model which has been used widely in previous scientific work and IPCC assessments. BernSCM represents the carbon cycle and climate system with a small set of equations for the heat and carbon budget, the parametrization of major nonlinearities, and the substitution of complex component systems with impulse response functions (IRF). The IRF approach allows cost-efficient yet accurate substitution of detailed parent models of climate system components with near linear behavior. Illustrative simulations of scenarios from previous multi-model studies show that BernSCM is broadly representative of the range of the climate-carbon cycle response simulated by more complex and detailed models. Model code (in Fortran) was written from scratch with transparency and extensibility in mind, and is provided as open source. BernSCM makes scientifically sound carbon cycle-climate modeling available for many applications. Supporting up to decadal timesteps with high accuracy, it is suitable for studies with high computational load, and for coupling with, e.g., Integrated Assessment Models (IAM). Further applications include climate risk assessment in a business, public, or educational context, and the estimation of $CO_2$ and climate benefits of emission mitigation options.

## 1   Introduction

Simple climate models (SCM) consist of a small number of equations, which describe the climate system in a spatially and temporally highly aggregated form. SCMs have been used since the pioneering days of computational climate science, to analyze the planetary heat balance (Budyko, 1969; Sellers, 1969), and to clarify the role of the ocean and land compartments in the climate response to anthropogenic forcing through carbon and heat uptake (e.g., Oeschger et al., 1975; Siegenthaler and Oeschger, 1984b; Hansen et al., 1984). Due to their modest computational demands, SCMs enabled pioneering research using the limited computational resources of the time, and continue to play a useful role in the hierarchy of climate models today.

Recent applications of SCMs are often found in research where computational resources are still limiting. Examples include probabilistic or optimization studies involving a large number of simulations, or the use of a climate component as part of a detailed interdisciplinary model. SCMs are also much easier to understand and handle than large climate models, which makes

them useful as practical tools that can be used by non-climate experts for applications where detailed spatio-temporal physical modeling is not essential. This applies to interdisciplinary research, educational applications, or the quantification of the impact of emission reductions on climate change.

An important application of SCMs is in Integrated Assessment Models (IAMs). IAMs are interdisciplinary models that couple a climate component with an energy-economy model, to simulate emissions and their climate consequences. Another application of simple models (e.g., Boucher and Reddy, 2008; Bruckner et al., 2003; Enting et al., 1994; Good et al., 2011; Hooss et al., 2001b; Huntingford et al., 2010; Joos and Bruno, 1996; Oeschger et al., 1975; Raupach, 2013; Siegenthaler and Oeschger, 1984a; Smith et al., 2017; Tanaka et al., 2007; Urban and Keller, 2010; Wigley and Raper, 1992) is to compare, analyze or emulate more complex models (Geoffroy et al., 2012b, a; Meinshausen et al., 2011; Raper et al., 2001; Thompson and Randerson, 1999). Simple models also play a significant role in previous assessments of the Intergovernmental Panel on Climate Change (e.g., Harvey et al., 1997). The comprehensive scope and interdisciplinarity of such models raise the challenge of maintaining a high and balanced scientific standard across all model components, especially when human resources are limited. This may apply particularly to the climate component, as IAMs are mostly used within the economic and engineering disciplines. Climate and carbon cycle representation are central parts of an IAM and have been critically assessed in the literature (Joos et al., 1999a; Schultz and Kasting, 1997; Vuuren et al., 2009).

BernSCM is a zero-dimensional global carbon cycle-climate model built around impulse-response representations of the ocean and land compartments, as described previously in Joos et al. (1996); Meyer et al. (1999). The linear response of more complex ocean and land biosphere models with detailed process descriptions is captured using impulse-response functions (IRFs). These IRF-based substitute models are combined with nonlinear parametrizations of carbon uptake by the surface ocean and the terrestrial biosphere as a function of atmospheric $CO_2$ concentration and global mean surface temperature. Pulse response models have been shown to accurately emulate spatially resolved, complex models (Joos et al., 1996; Joos and Bruno, 1996; Meyer et al., 1999; Joos et al., 2001; Hooss et al., 2001a).

BernSCM (Figure 1) is designed to compute decadal-to-millennial scale perturbations in atmospheric $CO_2$ in climate and in fluxes of carbon and heat relative to a reference state, typically preindustrial conditions. The uptake of excess, anthropogenic carbon from the atmosphere is described as a purely physico-chemical process (Prentice et al., 2001). As in pioneering modeling approaches with box-type (Oeschger et al., 1975; Revelle and Suess, 1957) and general ocean circulation models (Maier-Reimer and Hasselmann, 1987; Sarmiento et al., 1992) modification of the natural carbon cycle through potential changes in circulation and the marine biological cycle (Heinze et al., 2015) are not explicitly considered. While such modifications and their potential socio-economic consequences are vividly discussed in the literature (Gattuso et al., 2015), associated climate-$CO_2$ feedbacks are likely of secondary importance. Estimated uncertainties in the marine carbon uptake due to climate change, including warming-driven changes in $CO_2$ solubility, are found to be smaller in magnitude than uncertainties arising from imperfect knowledge of surface-to-deep physical transport (see Figure 2d,e in Friedlingstein et al., 2006b). The exchange of $CO_2$ between the atmosphere and the surface ocean is described by two-way fluxes, from the atmosphere to the surface ocean and vice versa, and the net flux of $CO_2$ into the ocean is proportional to the air-sea partial pressure difference. $CO_2$ reacts with water to form carbon and bicarbonate ions (Dickson et al., 2007; Orr et al., 2015), and acid-base equilibria are here described

using the well-established Revelle factor formalism (Siegenthaler and Joos, 1992; Zeebe and Wolf-Gladrow, 2001). The first order climate-carbon feedback of a decreasing solubility in warming water is considered. Surface-to-deep exchange, the rate limiting step of ocean carbon and heat uptake, is described using an IRF. On time scales of up to a few millennia, processes associated with ocean sediments and weathering can be neglected. In such a closed ocean-atmosphere-land biosphere system, excess $CO_2$ is partitioned between the ocean and the atmosphere and a substantial fraction of the emitted $CO_2$ remains in the atmosphere and in the surface ocean in a new equilibrium (Joos et al., 2013). This corresponds to a constant term (infinitely long removal time scale) in the IRF representing surface-to-deep mixing. On multi-millennial time scales, excess anthropogenic $CO_2$ is removed from the ocean-atmosphere-land system by ocean-sediment interactions and changes in the weathering cycle (Archer et al., 1999; Lord et al., 2016), and the IRF is readily adjusted to account for these processes, important for simulations extending over many millennia.

BernSCM simulates global mean surface temperature and the heat uptake by the planet. The latter is equivalent to the net top-of-the-atmosphere energy flux. Changes in the Earth's heat storage in response to anthropogenic forcing are dominated by warming of the surface ocean and the interior ocean (Stocker et al., 2013b) due to their large heat capacity in comparison with that of the atmosphere and their large thermal conductivity in comparison to that of the land surface. Consequently, the atmospheric and land surface heat capacity is formally lumped with the heat capacity of the surface ocean in the BernSCM. The uptake of heat by the ocean (or planet) is, as for carbon, formulated as a two-way exchange flux. The flux of heat from the atmosphere into the surface ocean is taken to be proportional to the radiative forcing resulting from changes in $CO_2$ and other agents (Etminan et al., 2016). The upward loss of heat from the surface is proportional to the product of the simulated surface temperature perturbation and the (prescribed) climate sensitivity $\lambda$ (Siegenthaler and Oeschger, 1984a; Winton et al., 2010).

As with carbon, surface-to-deep transport is the rate limiting step for ocean heat uptake and thus for the adjustment of surface temperature to radiative forcing. This transport is key to determine the lag between realized warming and equilibrium warming (Frölicher and Paynter, 2015). Again, this transport is described using an IRF. This IRF encapsulates the finite volume of the entire ocean. It also represents the range of transport time scales associated with advection, diffusion and convection ranging from decades for the ventilation of thermocline to more than a millennium for deep Pacific ventilation as evidenced by transient tracers such as CFCs and radiocarbon (Olsen et al., 2016). The simulated surface ocean temperature perturbation, taken as a measure of global mean surface air temperature change, may be combined with spatial patterns of change in temperature, precipitation or any other variable of interest to compute regionally explicit changes (Hooss et al. (2001b); Joos et al. (2001); Stocker et al. (2013a) (Figure 1).

Non-$CO_2$ radiative forcing may be prescribed, e.g., following estimates from complex climate-chemistry models (Myhre et al., 2013) or from simple emission driven non-$CO_2$ chemistry-radiative forcing modules (Joos et al., 2001; Smith et al., 2017) and reconstructions of solar and volcanic forcing (Eby et al., 2012; Jungclaus et al., 2017) and considering the forcing efficacy of non-$CO_2$ agents relative to $CO_2$ forcing (Hansen et al., 2005). Climate sensitivity characterizing the response to radiative forcing, is a free parameter in the BernSCM. Climate sensitivity may change under increasing warming, particularly in high emission scenarios (Geoffroy et al., 2012a; Gregory et al., 2015; Pfister and Stocker, 2017). Here, climate sensitivity is assumed to be time-invariant and a potential state dependency of climate sensitivity is not considered. This may be changed

when more solid information on state dependency becomes available or for the purpose of sensitivity analyses. Similarly, ocean heat uptake efficacy (Winton et al., 2010), influencing the atmospheric temperature response to ocean heat uptake forcing, is set to one here.

The present version 1.0 of BernSCM is fundamentally analogous to the Bern Model as used already in the IPCC Second Assessment Report, Bern-SAR (whereas different versions of the Bern model family were used in the more recent IPCC reports). BernSCM represents the relevant processes more completely than Bern-SAR, thanks to additional alternative representations of the land and ocean components, which contain a more complete set of relevant sensitivities to temperature and atmospheric $CO_2$.

Here, BernSCM model simulations are compared to previous multimodel studies. The model is run for an idealized atmospheric pulse $CO_2$ emission experiment of Joos et al. (2013); for an idealized $CO_2$ forcing experiment similar to simulations from the Climate Model Intercomparison Project 5 (CMIP5); and for the SRES A2 emission scenario used in the C4MIP study (Friedlingstein et al., 2006a).

Together with this publication, BernSCM v1.0 is provided as an open source Fortran code for free use. The code was also rewritten from scratch, with flexibility and transparency in mind. The model is comprehensively documented, and easily extensible. New alternative model components can be added using the existing ones as a template. A range of numerical solution schemes is implemented. Up to decadal timesteps are supported with high accuracy, suitable for the coupling with, e.g., emission models of coarse time resolution. However, the published code is a ready-to-run standalone model which may also be useful in its own right.

BernSCM offers a physically sound carbon cycle-climate representation, but it is small enough for use in IAMs and other computationally tasking applications. In particular, the support of long time steps is ideally suited to the application of BernSCM an IAM component, as these complex models often use time steps on the order of 10 years.

BernSCM also offers a tool to realistically assess the climate impact of carbon emissions or emission reductions and sinks, for example in aviation, forestry (Landry et al., 2016), blue carbon management, peat development (Mathijssen et al., 2017), life cycle assessments (Levasseur et al., 2016), or to assess the interaction of climate engineering interventions such as terrestrial carbon dioxide removal with the natural carbon cycle (Heck et al., 2016).

In this paper, we describe the model equations (section 2 and appendix B), illustrative simulations in comparison with previous multi-model studies and uncertainty assessment (section 3), followed by a discussion (section 4) and conclusions (section 5).

## 2 The BernSCM model framework and equations

BernSCM simulates the relation between $CO_2$ emissions, atmospheric $CO_2$, radiative forcing (RF), and global mean Surface Air Temperature (SAT) by budgeting carbon and heat fluxes globally between the atmosphere, the (abiotic) ocean, and the land biosphere compartments. Given $CO_2$ emissions and non-$CO_2$ RF, the model solves for atmospheric $CO_2$ and SAT (e.g., in

the examples of section 3), but can also solve for carbon emissions (or residual uptake) when atmospheric $CO_2$ (or SAT and non-$CO_2$ RF) is prescribed, or for RF when SAT is prescribed.

The transport of carbon and heat to the deep ocean, as well as the decay of land carbon result from complex, but in first order linear behavior of the ocean and land compartments. These are represented in BernSCM using impulse response functions (IRF, or Green's function). The IRF describes the evolution of a system variable after an initial perturbation, e.g., the pulse-like addition of carbon to a reservoir. It fully captures linear dynamics without representing the underlying physical processes (Joos et al., 1996). More illustratively, the ocean and land models can be considered to consist of systems of uncoupled first-order ordinary differential equations or "box models", which are an equivalent representation of the IRF model components (Figure 1).

The net primary production (NPP) of the land biosphere and the surface ocean carbon uptake depend on atmospheric $CO_2$ and surface temperature in a nonlinear way. These essential nonlinearities are described by parametrizations linking the linear model components.

## 2.1 Carbon cycle component

The budget equation for atmospheric carbon is

$$\frac{\mathrm{d}m_A}{\mathrm{d}t} = e - f_O - \frac{\mathrm{d}m_L}{\mathrm{d}t}, \tag{1}$$

where $m_A$ denotes the atmospheric carbon stored in $CO_2$, $e$ denotes $CO_2$ emissions, $f_O$ the flux to the ocean, $m_L$ the land biosphere carbon stock, and $t$ is time. Here, $m_L$ refers to the (potential) natural biosphere. Human impacts on the land biosphere exchange including land use and land use changes are not simulated in the present version, and treated as exogenous emissions ($e$). These emissions may be prescribed based on results from spatially-explicit terrestrial models. An overview of the model variables and parameters is given in tables A1 and A2.

The change in land carbon is given by the balance of net primary production (NPP) and decay of assimilated terrestrial carbon,

$$\frac{\mathrm{d}m_L}{\mathrm{d}t} = f_{\mathrm{NPP}} - f_{\mathrm{decay}} \tag{2}$$

Decay includes heterotrophic respiration (RH), fire and other disturbances due to natural processes.

Carbon is taken up by the ocean through the air-sea interface ($f_O$) and distributed to the mixed surface layer ($m_S$) and the deep ocean interior ($f_{\mathrm{deep}}$):

$$f_O = \frac{\mathrm{d}m_S}{\mathrm{d}t} + f_{\mathrm{deep}} \tag{3}$$

Global NPP ($f_{\mathrm{NPP}}$) is assumed to be a function of the partial pressure of atmospheric $CO_2$ ($p^{CO_2}$) and the SAT deviation from preindustrial equilibrium (functions for the implemented land components are given in Appendix A),

The net flux of carbon into the ocean is proportional to the gas transfer velocity ($k_g$) and the $CO_2$ partial pressure difference between surface air and seawater:

$$f_O = k_g A_O \, \varepsilon \, (p_A^{CO_2} - p_S^{CO_2}), \tag{4}$$

where $A_O$ is ocean surface area and $\varepsilon$ the atmospheric mass of C per mixing ratio of $CO_2$.

The global average perturbation in surface water $\Delta p_S^{CO_2}$ is a function of dissolved inorganic carbon change ($\Delta DIC$) in the surface ocean at constant alkalinity (Joos et al., 1996) and SAT (Takahashi et al., 1993); $\Delta DIC$ and $p_A^{CO_2}$ are related to model variables (see Appendix A),

$$\Delta DIC = \frac{m_S}{H_{mix} A_O \varrho M_{\mu mol} 10^{-15} Gt/g} \tag{5}$$

$$p_A^{CO_2} = m_A \varepsilon^{-1} \tag{6}$$

The carbon cycle equation set is closed by the specification of $f_{decay}$ and $f_{deep}$ (section 2.3), as well as $\Delta T$, i.e., the coupling to the climate component (section 2.2).

## 2.2 Climate component

BernSCM simulates the deviation in global mean SAT from the preindustrial state. SAT is approximated by the temperature perturbation of the surface ocean $\Delta T$, which is calculated from heat uptake by the budget equation

$$\frac{d\Delta T}{dt} c_S = f_O^H - f_{deep}^H, \tag{7}$$

where $c_s$ is the heat capacity of the surface layer, $f_O^H$ is ocean heat uptake, and $f_{deep}^H$ is heat uptake by the deep ocean (and accounts for the bulk of the effective heat capacity of the ocean). Continental heat uptake is neglected due to the much higher heat conductivity of the ocean in comparison to the continent.

$f_O^H$ is taken to be proportional to RF (Forster et al., 2007) and the deviation of SAT from radiative equilibrium ($\Delta T = \Delta T^{eq}(RF)$; see table A2 for parameter definitions),

$$f_O^H = RF \left(1 - \frac{\Delta T}{\Delta T^{eq}}\right) \frac{A_O}{a_O} \tag{8}$$

This relation follows from the assumption that feedbacks are linear in $\Delta T$ (e.g., Hansen et al., 1984). $\Delta T^{eq}$ is given by

$$\Delta T^{eq} = RF \frac{\Delta T_{2\times}}{RF_{2\times}}, \tag{9}$$

where $\Delta T_{2\times}$ is climate sensitivity (defined as the equilibrium temperature change corresponding to twice the preindustrial $CO_2$ concentration). Equation (8) describes ocean heat uptake as the difference between RF and the climate system's response, $\lambda \cdot \Delta T$, with $\lambda = RF/\Delta T^{eq}$ the climate sensitivity expressed in W m$^{-2}$ K$^{-1}$.

Climate sensitivity is an external parameter, as the model does not represent the processes determining equilibrium climate response. RF of $CO_2$ is calculated as (Myhre et al., 1998)

$$RF_{CO_2} = \ln\left(\frac{p_A^{CO_2}}{p_{A0}^{CO_2}}\right) \frac{RF_{2\times}}{\ln(2)}, \tag{10}$$

where $p_{A0}^{CO_2}$ is the preindustrial reference concentration of atmospheric $CO_2$, and $RF_{2\times}$ is the RF at twice the preindustrial $CO_2$ concentration. RF of other GHGs, aerosols etc. can be parametrized in similar expressions involving GHG and pollutant emissions and concentrations (Prather et al., 2001). In the provided BernSCM code, non-$CO_2$ RF is treated as an exogenous boundary condition. Total RF is then

$$RF = RF_{CO_2} + RF_{nonCO_2} \tag{11}$$

The calculation of $f_{deep}^H$ (section 2.3) completes the climate model.

## 2.3 Impulse response model components

The response of a time-invariant linear system to a time-dependent forcing $f$ can be expressed by

$$m(t) = \int_{-\infty}^{t} f(t') r(t - t') \, dt' \tag{12}$$

The function $r$ is the system's impulse response function (IRF), as can be shown by evaluating the integral for a Dirac impulse ($f(t') = \delta(t')$). The IRF indicates the fraction remaining in the system at time $t$ of a pulse input at a previous time $t'$. Because of linearity of the integral, any physically meaningful integrand $f$ can be represented as a sequence of such impulses of varying size.

In BernSCM, an IRF is used to calculate the perturbation of heat and carbon in the mixed surface ocean layer (mixed layer IRF, (Joos et al., 1996). For carbon,

$$m_S(t) = \int_{-\infty}^{t} f_O(t') r_O(t - t') \, dt', \tag{13}$$

and similarly, for heat

$$\Delta T(t) c_S = \int_{-\infty}^{t} f_O^H(t') r_O(t - t') \, dt' \tag{14}$$

This approach has been shown to faithfully reproduce atmospheric $CO_2$ and SAT as simulated with the models from which the IRF is derived (Joos et al., 1996). For temperature, the linear approach works since relatively small and homogeneous perturbations of ocean temperatures do not affect the circulation strongly and can be treated as a passive tracer (Hansen et al., 2010). Note that for compatibility with commonly used units, carbon fluxes are expressed in Gt per *year*, while heat fluxes are expressed in Joule per second (Watt) in equations 13 and 14, respectively.

Equation (13) closes the ocean C budget equation (3), as can be seen by taking the derivative with respect to time (using $r(0) = 1$),

$$\frac{dm_S}{dt} = f_O(t) - \underbrace{\left( - \int_{-\infty}^{t} f_O(t') \frac{dr_O}{dt}(t - t') \, dt' \right)}_{f_{deep}}, \tag{15}$$

where $f_{\text{deep}}$ is the flux to the deep ocean. Similarly, equation (14) closes the heat budget equation (7) for the surface ocean,

$$\frac{\mathrm{d}\Delta T}{\mathrm{d}t}c_S = f_O^H(t) - \underbrace{\left(-\int_{-\infty}^t f_O^H(t')\frac{\mathrm{d}r_O}{\mathrm{d}t}(t-t')\,\mathrm{d}t'\right)}_{f_{\text{deep}}^H} \tag{16}$$

Another IRF is used for the carbon $m_L$ in living or dead biomass reservoirs of the terrestrial biosphere,

$$m_L(t) = \int_{-\infty}^t f_{\text{NPP}}(t')r_L(t-t')\,\mathrm{d}t' \tag{17}$$

Again, equation (17) closes the budget equation for the land biosphere (2), as shown by the derivative with respect to time,

$$\frac{\mathrm{d}m_L}{\mathrm{d}t} = f_{\text{NPP}}(t) - \underbrace{\left(-\int_{-\infty}^t f_{\text{NPP}}(t')\frac{\mathrm{d}r_L}{\mathrm{d}t}(t-t')\,\mathrm{d}t'\right)}_{f_{\text{decay}}} \tag{18}$$

The time derivative of the land IRF is also known as the decay response function (e.g., Joos et al., 1996).

The above IRFs can be expressed as a sum of exponentials,

$$r(t) = a_\infty + \sum_k a_k e^{-t/\tau_k} \tag{19}$$

where the constant term $a_\infty$ corresponds to an infinite decay timescale.

The ocean IRF contains a positive constant coefficient $a_\infty$, indicating a fraction of the perturbation that will remain indefinitely (implied by carbon conservation in the ocean model). $CaCO_3$ compensation by sediment dissolution and weathering (Archer et al., 1999) are not considered here, but could be described using analogous elimination processes with time scales on the order of $10^4$ to $10^5$ yr (Joos et al., 2004). We emphasize that the implementation considering only the partitioning of
excess carbon between atmosphere, land and ocean (hence $a_\infty \neq 0$), neglecting ocean sediment-interactions and weathering flux perturbations, is only valid for time scales shorter than about 2,000 years. In land biosphere models, in contrast, organic carbon is lost to the atmosphere by oxidation to $CO_2$ at non-zero rates, and consequently all timescales are finite (i.e., $a_\infty = 0$), and the IRF tends to zero (Figure 2).

Inserting formula (19) in the pulse response equation (12) yields ($f$ is a perturbation flux when $a_\infty \neq 0$)

$$m(t) = \sum_k \int_{-\infty}^t f(t')\,a_k e^{-(t-t')/\tau_k}\,\mathrm{d}t' + \int_{-\infty}^t f(t')\,a_\infty\,\mathrm{d}t' \tag{20}$$

Thus the expression (12) separates into a set of independent integrals $m_k$ corresponding to the number of time scales of the response. Taking the time derivative of expression (20) reveals the equivalence to a system of uncoupled first-order ordinary differential equations

$$\frac{\mathrm{d}m_k}{\mathrm{d}t} = f(t)\,a_k - m_k/\tau_k; \qquad\qquad \frac{\mathrm{d}m_\infty}{\mathrm{d}t} = f(t)\,a_\infty$$

$$m = \sum_k m_k + m_\infty \tag{21}$$

The direct numerical evaluation of the equation (12) involves integrating over all previous times at each timestep. The differential form (21) allows a recursive solution, which is much more efficient, especially for long simulations (the recursive solution implemented in BernSCM is described in Appendix B).

The differential equation system (21) can be considered to consist of several boxes, whereby each box $m_k$ receives a fraction $a_k$ of the input $f$, and has a characteristic turnover time $\tau_k$ (Figure 1). In the following this is referred to as a "box model". For the mixed ocean surface layer the carbon content of box $k$ is given by:

$$\frac{dm_{S_k}}{dt} = f_O(t)\,a_{O_k} - m_{S_k}/\tau_{O_k}; \qquad\qquad \frac{dm_{S_\infty}}{dt} = f_O(t)\,a_{O_\infty} \tag{22}$$

and the change in total carbon content in the mixed layer is:

$$m_S = \sum_k m_{S_k} + m_{S_\infty} \tag{23}$$

Similar equations describe the heat content in the ocean surface layer, as well as the carbon stored in the land biosphere (Figure 1).

The timescales of an IRF describing a linear system are equivalent to the inverse eigenvalues of the model matrix of that system and may also be interpreted in the context of the Laplace transformation (Enting, 2007; Raupach, 2013). For example, the timescales of the mixing layer IRF are the inverse eigenvalues of a matrix describing a diffusive multilayer ocean model (Hooss et al., 2001a). A large model matrix yields a spectrum of many eigenvalues and timescales and corresponding model boxes. In practice, IRFs are approximated with fewer fitting parameters and, equivalently, timescales (4-6 in the case of BernSCM). Joos et al. (1996) used IRFs combined from two or more functions to minimize the number of parameters needed for an accurate representation. In BernSCM, simple IRFs of the form (19) are used exclusively. This allows adequate accuracy and a consistent interpretation as a multibox model.

Thinking of IRF components as box models is conceptually meaningful. The simple Bern 4 box biosphere model (Siegenthaler and Joos, 1992), for example, contains boxes corresponding to ground vegetation, wood, detritus, and soil (Appendix A). The High-Resolution Biosphere Model (HRBM) land component (Meyer et al., 1999), on the other hand, is abstractly defined by an IRF, but corresponds to boxes which correlate with biospheric reservoirs. However, since different box models may show a similar response, in practice the coefficients $a_k$ and time scales $\tau_k$ may not be uniquely defined by the IRF, and should be interpreted primarily as abstract fitting parameters (Enting, 2007; Li et al., 2009).

The IRF representation is, strictly speaking, only valid if the described subsystem is linear and the time scales of the system are time-invariant. Then, the response function $r$ does not depend on time and on state variables. In the BernSCM, major nonlinearities in the carbon cycle, namely air-sea gas exchange and the nonlinear carbonate chemistry and changes in NPP in response to changes in environmental conditions are treated by separate nonlinear equations (equations (4) and (5)), while surface-to-deep ocean transport of carbon and heat and respiration of carbon in litter and soils are viewed as approximately linear processes using IRFs. Yet ocean circulation and the respiration of carbon from soil and litter is likely to change under global warming, violating assumption of linearity. In practice, the IRF representation remains a useful approximation as long as the impact of associated nonlinearities on simulated atmospheric $CO_2$ and temperature remain moderate.

The interpretation of the IRF representation as a box model provides a starting point for considering nonlinearities in the response. To account for nonlinearities, the response time scales $\tau_k$ and the coefficients $a_k$ may be gradually adjusted as a function of state variables such as temperature. As the integral form (12) involves integration over the whole history at each time step, changing parameters along the way would result in inconsistencies. In contrast, the differential or box-model form (21) does not depend on previous time steps. Changing the model parameters from one step to the next thus equates to applying a slightly different model at each time step. Within each time step, the parameters remain constant, and the solution for the linear case applies. As time steps are small compared to the whole simulation, this discretization yields accurate results, which is confirmed by the close agreement between the different time resolutions (Table A5).

Varying coefficients have been successfully implemented and tested for the HRBM land component and its decay IRF (Meyer et al., 1999). In this way, the enhancement of biomass decay by global warming is captured (s.a. Appendix A and section 3.1). In such a modification, the advantage of the IRF and the equivalent box model representation - the faithful representation of the characteristic response time scale of a model system - is largely maintained, while at the same time the impact of time and state-dependent system responses on simulated outcomes is approximated.

## 3  Illustrative simulations with the BernSCM

### 3.1  Model setup for sensitivity analyses and uncertainty assessment

The carbon cycle-climate uncertainty of simulations with BernSCM can be assessed in two ways. First, to assess structural uncertainty, different substitute models for the ocean and land components can be used. Currently, this approach is quite limited by the set of available substitute models (see Appendix A). Second, parameter uncertainty can be assessed by varying the temperature and $CO_2$ sensitivities of the model, based on a standard set of components that represent the key dependencies as completely as possible (here, the IRF substitutes for the High-Latitude Exchange/Interior Diffusion-Advection (HILDA) ocean model (Joos et al., 1996) and for the HRBM land biosphere model (Meyer et al., 1999) are used in the standard setup).

The uncertainties of the global carbon cycle concern the sensitivity of the modeled fluxes of carbon and heat to changing atmospheric $CO_2$ and climate. Key uncertainties strongly affecting the overall climate response are associated with land C storage: the dependency of NPP on $CO_2$ ($CO_2$ fertilization), and the dependency of land C on temperature ($f_{\text{decay}}$ increases with warming). This gives rise to large and opposed carbon flux perturbations which are both very uncertain in magnitude (Le Quéré et al., 2016). While all substitute land models available for BernSCM include $CO_2$ fertilization, only the HRBM substitute model represents temperature sensitivity of biomass decay (Appendix A2).

As for the ocean, the uncertainty of heat uptake into the surface ocean is treated in terms of climate sensitivity (eq. 8). The efficiency of the uptake of heat ($f_{\text{deep}}^H$) and carbon ($f_{\text{deep}}$) into the deep ocean is not sensitive to temperature, as the currently available substitute models all represent a fixed circulation pattern (IRF/box-model parameters are not temperature dependent, Appendix A1). The nonlinear chemistry of $CO_2$ dissolution in the surface ocean (eq. 4), which determines the sensitivity of ocean C uptake to atmospheric $CO_2$, is scientifically well established (Dickson et al., 2007; Orr and Epitalon, 2015), and is

not treated as an uncertainty in BernSCM. The temperature sensitivities of NPP and $CO_2$ dissolution in the surface ocean are treated as uncertain here, but have secondary influence on the climate response.

Similar to previous studies using models from the Bern family (Plattner et al., 2008; Joos et al., 2001; Meehl et al., 2007; Van Vuuren et al., 2008), the parameter uncertainty range is assessed using the following setups:

**"coupled":** All temperature and $CO_2$ sensitivities at their standard values

  **"uncoupled":** All sensitivities zero (except from the ocean $CO_2$ dissolution chemistry)

  **"C-only":** Only $CO_2$ dependencies considered ($CO_2$ fertilization)

  **"T-only":** Only temperature dependencies considered in land module (NPP, decay)

We performed simulations with these different setups. In section 4.2, we probe the time scales of the temperature response
in simulations where atmospheric $CO_2$ is abruptly (instantaneously) quadrupled or by increasing $CO_2$ radiative forcing linearly within 140 years. In section 4.3, we probe the response of the coupled system to a pulse-like release of 100 GtC into the atmosphere. Finally in section 4.4, we analyze carbon cycle-climate feedbacks relying on simulations over the industrial period and for the SRES A2 scenario. BernSCM results are compared with the results from three multi-model intercomparison projects: the Climate Model Intercomparison Project 5 (CMIP5) with results as summarized by Frölicher and Paynter (2015);
an analysis of carbon dioxide and climate impulse response functions (Joos et al., 2013, here referred to as IRFMIP), and the C4MIP Climate-Carbon Cycle Feedback Analysis (Friedlingstein et al., 2006a).

### 3.2   Fraction of realized warming and idealized forcing experiments

The climate response of BernSCM is illustrated using idealized simulations with prescribed forcing. One series of simulations (a) was run for $CO_2$ concentration increasing exponentially from the preindustrial value by 1% per year over 140 years to
approximately four times the preindustrial concentration, corresponding to a linear increase in RF (Figure 3, panel a); in a second series of simulations (b), $CO_2$ was abruptly increased to four times the preindustrial concentration (Figure 3, panel b).

Frölicher and Paynter (2015) compare similar simulations of Earth System Models (ESM) performed within the Coupled Model Intercomparison Project Phase 5 (CMIP5), and Earth System Models of Intermediate Complexity (EMIC) (Joos et al., 2013). As a model comparison metric sensitive to the long-term climate response, Frölicher and Paynter (2015) use the fraction
of realized warming, defined by the ratio of the temperature response at a given year and the equilibrium temperature for the corresponding RF. They show that the smaller realized warming of ESMs in comparison to EMICs (Figure 3) is connected to a higher long-term warming response; this implies an increase in the coefficient relating global warming to cumulative carbon emissions on multi-centennial timescales and suggests a lower quota on allowed emissions for a given global warming target (Frölicher and Paynter, 2015). The realized warming fraction simulated with BernSCM is in good agreement with the
responses of the ESMs (and lower on average than that of the EMICs). The validity of the IRF approach has also been shown by Good et al. (2011) using a SCM to reconstruct and interpret AOGCM projections. For the 150-year time scale of the CMIP5

experiments, Geoffroy et al. (2012b, a) show that the climate response of AOGCMs is well captured by a two-layer energy balance model with two effective response time scales.

In BernSCM, the fraction of realized warming depends primarily on the choice of climate sensitivity, and is qualitatively similar for the different model setups. Such a clear relationship is not seen in the EMS and EMICS. Thus the structural uncertainty and model differences of complex models is not fully represented in BernSCM. The BernSCM climate response to abrupt warming (Figure 3, panel b) is qualitatively similar, especially on multi-centennial time scales.

### 3.3 Impulse response experiment

Coupled carbon cycle-climate models can be characterized and compared based on their response to a $CO_2$ emission pulse to the atmosphere (Joos et al., 2013). The airborne fraction (AF) denotes the fraction of emissions found in the atmosphere at at a given time. In IRFMIP, the AF for a pulse of $100\,\text{GtC}$, emitted on top of current (i.e., year 2010) atmospheric $CO_2$ concentrations, was simulated by a set of 15 carbon cycle-climate models of different complexity. For three of these models (Bern3D-LPJ, GENIE, MAGICC), ensembles sampling the parameter uncertainty of these models are included in IRFMIP. Thus, IRFMIP captures structural as well as parameter uncertainty.

The IRFMIP pulse experiment was repeated with BernSCM, exploring parameter uncertainty of the carbon cycle (section 3.1), as well as structural uncertainty, using the ocean model IRFs HILDA and Princeton (Sarmiento et al., 1992) in various combinations with the land biosphere components HRBM and Bern-4box (Figure 4). Simulations were run for equilibrium climate sensitivities of 3°C (standard setup), 2°C, and 4.5°C.

The AF simulated with BernSCM broadly agrees with the set of simulations from IRFMIP. 100 years after the pulse, it is 0.40 (0.34–0.57) for a climate sensitivity of 3°C (for coupled setup with uncertainty range in brackets). Climate sensitivity uncertainty only slightly affects the upper end of this range (Figure 4). For AF simulated with BernSCM, the standard coupled setup is close to the IRFMIP multimodel median. The BernSCM uncertainty range is asymmetric, like the IRFMIP multimodel range. For the MAGICC and GENIE ensembles, the medians also correspond with the BernSCM standard case, while the uncertainty ranges are more symmetric.

The BernSCM SAT response also broadly agrees with IRFMIP. The standard coupled simulation is somewhat lower than the IRFMIP median, which is explained in part by the climate sensitivity (3°C) being slightly lower than the IRFMIP average (3.2°C). The short term temperature response of BernSCM in particular is on the lower side of the IRFMIP range, suggesting stronger ocean mixing. The quickest initial temperature increase of the BernSCM simulations is obtained with the Princeton ocean model component (dashed lines), which shows a slower initial mixing to the deep ocean than the other implemented components (Figure 2). The comparability of the SAT projections is limited, as the range of climate sensitivities considered in the BernSCM simulations (2-4.5°C) differ somewhat from that of the IRFMIP multimodel set (1.5-4.6°C) and the single model ensembles (1.9-5.7°C), and are compounded with RF differences resulting from the uncertainty in atmospheric $CO_2$.

Figure 5 shows how the added carbon is redistributed within the Earth system. In the coupled setup, the fraction of the initial pulse sequestered by the land and by the ocean increases over the first century, while the airborne fraction decreases. After 100 years, slightly more than 20% of the added carbon is stored in the land and about 40% in the ocean. The ocean continues to

sequester excess carbon in the following centuries to become the dominant sink for excess carbon. In contrast, the land returns part of the sequestered carbon back to the atmosphere and ocean as decreasing atmospheric $CO_2$ is reducing the modeled $CO_2$ fertilization of the land biosphere. In the T-only setup, where $CO_2$ fertilization is not operating, the land is a source of carbon to the atmosphere due to accelerated soil turnover in response to warming. The largest land sink is simulated in the C-only

setup, where soil turnover timescales remain invariant and $CO_2$ fertilization is on. The different BernSCM setups span a range of plausible land biosphere and ocean responses to continued anthropogenic $CO_2$ emissions as reflected in the simulated range in the airborne fraction (Figure 4a, 5)."

## 3.4 Carbon cycle-climate feedbacks

Climate models with explicit and detailed carbon cycle components exhibit a wide range of responses, as shown in the inter-

comparison studies of climate models with a detailed carbon cycle, C4MIP (Friedlingstein et al., 2006a) and CMIP5 (Jones et al., 2013). The authors analyzed the feedback of carbon cycle-climate models using linearized sensitivity measures. These are derived from a simulation with temperature dependence ("coupled") and one without ("uncoupled"; note that these names have a different meaning in BernSCM). Total $CO_2$ emissions for the "coupled" (left hand side) and "uncoupled" (right hand side) simulations can be expressed as

$$\Delta C_A^c(\varepsilon + \beta_L + \beta_O + \alpha(\gamma_L + \gamma_O)) = \Delta C_A^u(\varepsilon + \beta_L + \beta_O) \qquad (24)$$

where $\Delta C_A$ is the cumulative change in atmospheric $CO_2$ (in ppm) in the coupled ($c$) or uncoupled ($u$) case, and the terms in parentheses represent the total sensitivity of C storage to $\Delta C_A$; in particular, $\beta$ is the sensitivity of carbon storage to atmospheric $CO_2$ (in GtC/ppm) on land ($\beta_L$) or in the ocean ($\beta_O$). $\gamma$ is similarly the sensitivity in carbon storage to climate change, and $\alpha$ is the linear transient climate sensitivity to $CO_2$ (°C per ppm) as in Friedlingstein et al. (2006b); $\varepsilon$ converts ppm

to GtC (cf. Table A2; the formula in the original paper implies identical units for atmospheric and stored carbon).

The climate-carbon cycle feedback is measured by the feedback metric $g$, defined by

$$\frac{\Delta C_A^c}{\Delta C_A^u} = \frac{1}{1 - g} \qquad (25)$$

and is thus estimated by

$$g = -\frac{\alpha(\gamma_L + \gamma_O)}{\varepsilon + \beta_O + \beta_L} \qquad (26)$$

Thus the feedback strength scales with the assumed climate sensitivity and the temperature sensitivities, and is reduced by $CO_2$-induced sinks.

The C4MIP study used a SRES A2 emission scenario to compare the carbon cycle sensitivities of a range of models. As in the C4MIP exercise, BernSCM was run for SRES A2 without any non-$CO_2$ forcings (Figure 6; prescribed historical and scenario emissions were smoothed with the R smooth.spline function (R Core Team, 2015) for 41 degrees of freedom for use

with different time steps). Land use was treated as an exogenous $CO_2$ emission, while the land model simulates an undisturbed biosphere.

The BernSCM sensitivity setups can be expressed in terms of the C4MIP sensitivity metrics: T-only corresponds to $\beta_L = 0$, C-only to $\gamma_L = \gamma_O = 0$, and uncoupled to $\beta_L = \gamma_L = \gamma_O = 0$. This can be used to estimate climate-carbon cycle feedback $g$ captured in BernSCM. The sensitivity metrics for the BernSCM standard simulation (HILDA-HRBM with coupled carbon cycle) lie within the C4MIP range (Table 1). The uncertainty range for BernSCM, however, is not congruent with the multi-model range of C4MIP. Maximum and standard sensitivity for BernSCM are practically identical. Notably, this sensitivity is smaller (absolutely) than the C4MIP average for the land carbon response to $CO_2$ increase and warming. The resulting gain $g$ is also smaller, though this results in large part from the lower climate sensitivity in BernSCM (which corresponds to 2.5°C as used for the Bern-CC model contribution to C4MIP). The lower end (in absolute terms) of the BernSCM carbon cycle sensitivity range is, on the other hand, zero per definition for all but the ocean-$CO_2$ sensitivity $\beta_O$ (see section 3.1). As a consequence, the climate-carbon cycle feedback range also includes zero. In contrast, the C4MIP range does not include zero for all sensitivity parameters.

The land carbon uptake until 2100, under the different BernSCM configurations, varies over 500 GtC (Figure 6), more than three times the range of ocean uptake (180 GtC). This partly reflects the limited coverage of the uncertainty in ocean mixing, but also the fact that the land carbon sink is, together with the land use-related source, the most uncertain item in the budget (Le Quéré et al., 2009).

## 4    Discussion

We simulated illustrative scenarios from two recent multi-model studies, C4MIP and IRFMIP, to compare BernSCM to the literature of carbon-cycle climate models. The results show that BernSCM is broadly representative of the current understanding of the global carbon cycle-climate response to anthropogenic forcing (in a time-averaged sense that does not address internal variability). The BernSCM uncertainty range in $CO_2$ and SAT projections is broadly similar to the ranges spanned by probabilistic single-model ensembles, and multi-model "ensembles of opportunity" such as the 15 IRFMIP models. The shown BernSCM uncertainty range consists mainly of parameter uncertainty and to a small extent of structural uncertainty. For the standard, coupled model setup, the sensitivities of ocean and land carbon uptake to changing $CO_2$ and climate (Table 1) of BernSCM are within the range of the detailed carbon cycle models in C4MIP. However, as some C4MIP models show much higher sensitivities, the BernSCM range does not capture the full C4MIP multi-model range. On the other hand, the C4MIP set is unlikely to sample uncertainty exhaustively, as each model contributed only a single, "most likely" simulation. Thus it does not include zero (or weak) sensitivities, whereas the BernSCM range does.

As Figure 6 shows, solutions with different timesteps and numerical schemes as implemented in BernSCM are largely equivalent for a sufficiently smooth forcing. This offers the flexibility to opt for simplicity of implementation or maximum speed as required by the application (see also Appendix B).

BernSCM does not explicitly distinguish between surface atmosphere and surface ocean temperature to compute global mean surface air temperature perturbation. This is in contrast to some energy balance calculations used to analyze results from state-of-the-art Earth System Models (e.g., Geoffroy et al., 2012b). The BernSCM approach follows earlier work of

Siegenthaler and Oeschger (1984a). It is further guided by the similarity in reconstructions of marine night time air and sea surface temperature perturbations (Stocker et al., 2013b) that are consistent with the short, monthly relaxation time scale for air-sea heat exchange. The focus of the BernSCM is on the representation of the transport of heat from the surface into the thermocline and the deep ocean on decadal to multi-century time scales, while information on seasonal and spatial changes such as on land-sea air temperature differences or polar amplification may be obtained by applying suitable spatial perturbation patterns as derived from state-of-the-art models.

Currently, a limited set of substitute models is available and included with BernSCM. The simple structure and open source policy of BernSCM allows users to address these current limitations according to the needs of their applications. More components can be added using the existing ones as a template. This requires the specification of the IRF and the parametrization of gas exchange for the surface ocean, or NPP for the land biosphere, respectively (as described in Joos et al., 1996; Meyer et al., 1999).

Ocean transport is known to vary under climate change with some consequences for heat and carbon uptake (Joos et al., 1999b). Here, we applied time-invariant ocean transport parameters ($a_{O_k}$, $\tau_{O_k}$). It is in principle possible to represent temperature dependency of ocean transport in a similar way as it is done for the climate dependency of heterotrophic respiration for the HRBM land biosphere substitute model (Meyer et al., 1999). In the current BernSCM version, the same IRF parameters are applied for the transport of carbon and heat from the surface ocean to the interior ocean. Thereby, it is implicitly assumed that the spatial pattern of change is the same for temperature and carbon. This appears to be a reasonable first-order approximation on decadal-to-century time scales as perturbations in temperature and carbon show similar patterns with decreasing perturbations from the surface to depth. In future efforts, one may differentiate the ocean IRF for heat and carbon, in particular when more information from long-term multi-century to millennial-scale ESM simulations becomes available. The application of the same IRF for carbon and heat in individual model runs implies that modeled carbon and heat transport tend to be physically consistent. In contrast, some other simple models employ different transport parameters for heat and carbon and varied these parameters independently in probabilistic studies.

A distribution of time scales applies to ocean transport processes as evidenced by observations of transient and time dependent tracers such as chlorofluorocarbons, bomb-produced and natural radiocarbon and biogeochemical tracers (Key et al., 2004; Olsen et al., 2016). This continuum is sometimes approximated by one time scale, also termed heat uptake efficiency (e.g., Gregory et al., 2009) and by two time scales, as in (Geoffroy et al., 2012b). The one-to-two time scale approximations were used to analyze relatively short Earth System Model simulations that do not yet reveal the multi-century response time scales of the deep ocean. We note that the equivalent ocean depth of the simple energy balance model of Geoffroy et al. (2012b) for their AOGCM ensemble is only 1,182 m compared to a mean ocean depth of about 3,800 m. The ocean IRFs used in the BernSCM are derived from long simulations with ocean-only or simplified models. The range of distinct time scales used to construct the IRF faithfully approximates the sub-annual to multi-century response continuum of the parent models as shown in earlier work (Joos et al., 1996). Further, the BernSCM IRF model represents the heat capacity of the entire ocean.

The BernSCM model may be extended to model perturbation in the signatures and exchange fluxes of the carbon isotopes $^{13}$C and $^{14}$C as demonstrated in earlier work (Joos et al., 1996). This was not implemented here to keep the code as simple as possible and as most potential users are likely concerned with the evolution of climate and atmospheric $CO_2$ .

A potential future application of BernSCM is to use it as an emulator of the global long-term response of complex climate-carbon cycle models by adding the corresponding substitute model components. Additionally, pattern scaling can be applied to transfer the global mean temperature signal into spatially resolved changes in surface temperature, precipitation, cloud cover, etc., exploiting the correlation of global SAT with regional and local changes (Hooss et al., 2001a). This allows to drive spatially explicit models, e.g., of terrestrial vegetation (as in Joos et al., 2001; Strassmann et al., 2008) or climate change-related impacts (e.g., as in Hijioka et al., 2009). Patterns of change are generally similar across models for temperature, whereas patterns in precipitation are more uncertain and show greater variability between models (Knutti and Sedlacek, 2013) and are forcing dependent (Shine et al., 2015). We also note that natural variability strongly influences the space-time evolution of climate change (Deser et al., 2012). Patterns may be scaled with changes in global mean surface air temperature as indicated in Figure 1 or dependencies on radiative forcing may be considered (Shine et al., 2015)

The addition of further alternative model components will extend the structural uncertainty that can be represented with BernSCM. A sufficient coverage of structural uncertainty could allow the interpolation between alternative model components, to represent uncertainty with scalable parameters (and removing the distinction between structural and parameter uncertainty). Such a parametrization of the uncertainty would enhance the possibilities for probabilistic applications of BernSCM, although more sophisticated models are available for observation-constrained probabilistic quantification of climate targets (Holden et al., 2010; Steinacher and Joos, 2016; Steinacher et al., 2013).

## 5    Conclusions

BernSCM is a reduced-form carbon cycle-climate model that captures the characteristics of the natural carbon cycle and the climate system essential for simulating the global long term response to anthropogenic forcing. Simulated atmospheric $CO_2$ concentrations and SAT are in good agreement with results from two comprehensive multi-model ensembles. Process detail is minimal, due to the use of IRFs for system compartments that can be described linearly, and nonlinear parametrizations governing the carbon fluxes into these compartments. This framework allows, in particular, to represent the wide range of response time scales of the ocean and land biosphere, and the nonlinear chemistry of $CO_2$ uptake in the surface ocean - both essential for reliably simulating the global climate response to arbitrary forcing scenarios.

Due to its structural simplicity and computational efficiency, BernSCM has many potential applications. In combination with pattern scaling, BernSCM can be used to project spatial fields of impact-relevant variables for applications such as climate change impact assessment, coupling with spatially explicit land biosphere models, etc. With alternative numerical solutions of varying complexity and stability to choose from, applications range from educational to computationally intensive integrated assessment modeling. BernSCM also offers a model-based alternative to GWPs for estimation of the climate impact of

emissions and can be used to quantify climate benefits of mitigation options by applying emissions- or concentration-driven simulations.

The generic implementation of linear IRF-components offers a transparent, extensible climate model framework. Current limitations concern the number of available substitute models (limiting the uncertainty range represented), and ocean transport not influenced by climate change. An addition of further alternative model components, and more flexible representation of sensitivities in terms of continuously variable parameters would further increase the models usefulness, for example for probabilistic applications.

*Code availability.* The source code of the Bern Simple Climate Model is available from the github repository at https://doi.org/10.5281/zenodo.1038117

## Appendix A: Model parameters and parametrizations

### A1 Ocean

Currently available ocean components include substitute models for the High-Latitude Exchange/Interior Diffusion-Advection model (HILDA Joos et al., 1996), Bern2D (Stocker et al., 1992), and the Princeton GCM (Sarmiento et al., 1992). Ocean model parameters of the equations described in the main text are listed in Table A3 for the mixed-layer IRF/box models and in Table A2 for other equations. The IRF/box model parameters given here are recalculated by fitting a sum of 6 exponential functions and one constant to the original response functions as given in Joos and Bruno (1996). The original functions treated the first few years separately; the approximation to a purely exponential form simplifies the equations and has a negligible effect on accuracy. The parametrization of ocean surface $CO_2$ pressure is the same for all available ocean components and is given below.

Ocean surface $CO_2$ pressure perturbations are fitted as a function of the globally averaged unperturbed surface temperature $T^*$ and perturbations in DIC by Joos et al. (1996) using carbonate chemistry coefficients summarized by Millero (1995):

$$\Delta\,p_S^{CO_2}\Big|_{T^*} = (1.5568 - 1.3993 \cdot 10^{-2}\,T^*)\,\Delta DIC + (7.4706 - 0.20207\,T^*)\,10^{-3}\,\Delta DIC^2 - (1.2748 - 0.12015\,T^*)\,10^{-5}\,\Delta DIC^3$$
$$+ (2.4491 - 0.12639\,T^*)\,10^{-7}\,\Delta DIC^4 - (1.5468 - 0.15326\,T^*)\,10^{-10}\,\Delta DIC^5$$

The expression holds for unperturbed global average surface water temperature $T^*$ between 17.7 and 18.3°C and for $\Delta p_S^{CO_2}$ between 0 and 1320 ppm.

Ocean surface $CO_2$ pressure for global surface temperature perturbation $\Delta T$ (Takahashi et al., 1993):

$$p_S^{CO_2} = p_S^{CO_2}\Big|_{T^*} \cdot e^{0.0423\Delta T}$$

### A2 Land biosphere

Currently available land biosphere components include substitute models for the High-Resolution Biosphere Model (Meyer et al., 1999) and the 4Box biosphere model (Siegenthaler and Joos, 1992).

For the HRBM model, temperature-dependent IRF/box model parameters as given by Meyer et al. (1999) are implemented:

$$\tilde{a}_k = \frac{a_k \, e^{s_{a_k} T}}{\sum_j a_j \, e^{s_{a_j} T}},$$

$$\tilde{\tau}_k = \tau_k \, e^{-s_{\tau_k} T},$$

where $\tilde{a}_k$, $\tilde{\tau}_k$ are the adjusted and $a_k$, $\tau_k$ the unperturbed parameters. The IRF/box model parameter values for HRBM and the 4box model are listed in Table A4. The temperature sensitivities of the HRBM IRF are parametrized for a warming of up to

$5°C$.

Net primary production for HRBM is given by (Meyer et al., 1999):

$$\text{NPP}(p)|_{\Delta T=0} = -e^{3.672801} + e^{-0.430818} \cdot p - e^{-6.145559} \cdot p^2 + e^{-12.353878} \cdot p^3 - e^{-19.010800} \cdot p^4 + e^{-26.183752} \cdot p^5$$
$$- e^{-34.317488} \cdot p^6 - e^{-41.553715} \cdot p^7 + e^{-48.265138} \cdot p^8 - e^{-56.056095} \cdot p^9 + e^{-64.818185} \cdot p^{10}$$

where $p$ is atmospheric $CO_2$ pressure. This expression holds up to a $CO_2$ concentration of 1274 ppm and is capped at that value. The model includes growth enhancement by SAT increase (but without a dynamical vegetation):

$\text{NPP}(p, \Delta T) = \text{NPP}_0 \cdot (1 + 0.11780208 \tanh(\Delta T / 50.9312421) + 0.002430513 \cdot \tanh(\Delta T / 8.85326739))$

This expression is holds up to a SAT increase of $5°C$.

Net primary production for the 4Box model is described after (Enting et al., 1994; Schimel et al., 1996):

$$\text{NPP} = \text{NPP}_0 + \text{NPP}_0 * \beta * \log(p^{CO_2} / p_0^{CO_2})$$

where $\text{NPP}_0$ is undisturbed NPP.

## Appendix B: Implementation of the pulse-response model

### B1    Discretization

For the solution of the pulse-response equation (12), two discrete approximations are implemented, using the separation by time scales in equation (20) or, equivalently, in the differential equation system (21). The recursive solution for a time step $\Delta t$ can be obtained from equation (20) by substituting $t = t_n = t_{n-1} + \Delta t$, and $s = t' - t_{n-1}$,

$$m_n = m_{\infty n} + \sum_k m_{kn}$$

$$m_{kn} = m_{kn-1} e^{-\Delta t / \tau_k} + \int_0^{\Delta t} f(t_{n-1} + s) \, a_k e^{-(\Delta t - s)/\tau_k} \, \mathrm{d}s$$

$$m_{\infty n} = m_{\infty n-1} + \int_0^{\Delta t} f(t_{n-1} + s) \, a_\infty \, \mathrm{d}s$$

(B1)

where $m_n = m(t_n) = m(t_{n-1} + \Delta t)$.

First, $f$ can be taken as constant over a sufficiently short timestep $\Delta t = t_i - t_{i-1}$. Evaluating equations (B1) yields

$$m_{kn} = m_{kn-1}\, e^{-\Delta t/\tau_k} + f(t^*)\, a_k \tau_k (1 - e^{-\Delta t/\tau_k})$$

$$m_{\infty n} = m_{\infty n-1} + f(t^*)\, a_\infty \Delta t \tag{B2}$$

where $t^*$ is chosen to be $t_{n-1}$ (for explicit forward solution) or $t_n$ (for implicit backward solution).

Second, for longer timesteps, a better approximation is obtained by assuming linear variation of $f$ over each time step. This yields

$$m_{kn} = m_{kn-1}\, e^{-\Delta t/\tau_k} + f_{n-1} a_k \tau_k \left( \frac{\tau_k}{\Delta t}(1 - e^{-\Delta t/\tau_k}) - e^{-\Delta t/\tau_k} \right) + f_n a_k \tau_k \left( 1 - \frac{\tau_k}{\Delta t}(1 - e^{-\Delta t/\tau_k}) \right)$$

$$m_{\infty n} = m_{\infty n-1} + \frac{f_{n-1} + f_n}{2} a_\infty \Delta t \tag{B3}$$

## B2 Numerical schemes

For the solution of the BernSCM model equations, both explicit and implicit time stepping is implemented.

The stability requirement for the numerical solution depends on the equilibration time for the ocean surface $CO_2$ pressure $p_S^{CO_2}$. Due to the buffering of the carbonate chemistry, the $CO_2$ equilibration time is smaller than the gas diffusion time scale ($\sim 10\,\mathrm{yr}$) by a ratio given by the buffer factor. For undisturbed conditions (buffer factor $\simeq 10$) the equilibration time is about $1\,\mathrm{yr}$. With increasing DIC, the buffer factor increases and the equilibration time shortens, making the equation system stiffer. Accordingly, when the model is solved explicitly with a time step of $1\,\mathrm{yr}$, instability typically occurs after sustained carbon

uptake by the ocean, which can occur in many realistic scenarios.

For the tested scenario range, the explicit solution is stable at a time step on the order of $0.1\,\mathrm{yr}$, for which the piecewise constant approximation is accurate. For larger step size, an implicit solution is required to guarantee stability.

The piecewise constant approximation is adequate for time steps up to $1\,\mathrm{yr}$, and the piecewise linear approximation for up to decadal time steps. An overview of the performance of three representative settings (set at compile time) for the C4MIP A2

scenario is given in Table A5.

The explicit solution is only implemented for the piecewise constant approximation (B2) and the implicit solution for both the piecewise constant (B2) and the piecewise linear approximation (B3). Equations (B2,B3) are expressed in a common equation by substituting

$$m_{kn} = m_{kn-1} p_{mk} + f_n p_{fk} + f_{n-1} p_{fk}^{\mathrm{old}} \tag{B4}$$

In the following, the implicit solution for the piecewise constant discretization is derived. Here, the fully implicit scheme for land and ocean exchange is discussed, but for stability, it is only crucial to treat ocean uptake implicitly. The parameters of equation (B4) for this case are

$$p_{mk} = e^{-\Delta t/\tau_k}$$

$$p_{fk} = a_k \tau_k (1 - e^{-\Delta t/\tau_k})$$

$$p_{fk}^{\mathrm{old}} = 0 \tag{B5}$$

Consider first the equation system for carbon, assuming temperature to be known (or neglecting temperature dependence of model coefficients). Equation (B4) is applied to land carbon exchange for the constant approximation (B5),

$$m_{Ln} = m_L^{c\Delta} + \Delta f_{\text{NPP}} \sum_k p_{fk\,L}$$

$$m_L^{c\Delta} = \sum_k m_{Lkn-1} p_{mk\,L} + f_{\text{NPP}\,n-1} \sum_k p_{fk\,L} \tag{B6}$$

where $m_L^{c\Delta}$ is the land carbon stock obtained after one time step if NPP remained constant ("constant flux commitment"), and

$\Delta f_{\text{NPP}} = (f_{\text{NPP}\,n} - f_{\text{NPP}\,n-1})$ is the change in NPP over one time step.

For ocean carbon uptake,

$$m_{Sn} = m_S^{c0} + f_{O\,n} \sum_k p_{fk\,O}$$

$$m_S^{c0} = \sum_k m_{Skn-1} p_{mk\,O} \tag{B7}$$

where $m_S^{c0}$ is the value of $m_S$ after one time step if $f_{O\,n} = 0$ ("zero-flux commitment").

To solve the implicit system, the nonlinear parametrizations need to be linearized around $t_{n-1}$. Linearizing ocean surface

$CO_2$ pressure as a function of surface ocean carbon and inserting in equation (4) yields

$$f_{O\,n} \simeq k_g A_O (m_{A\,n} - \varepsilon\, p_{S,n-1}^{CO_2}) + k_g A_O\, \varepsilon\, \frac{dp_S^{CO_2}}{dm_S}\bigg|_{n-1} (m_{S\,n-1} - m_{S\,n}) \tag{B8}$$

where equations (5,6) were used. Similarly, NPP as a function of atmospheric carbon is linearized,

$$\Delta f_{\text{NPP}\,n} \simeq \frac{df_{\text{NPP}}}{dm_A}\bigg|_{n-1} (m_{A\,n} - m_{A\,n-1}) \tag{B9}$$

using equation (6).

The system is completed with the discretized budget equation (1)

$$m_{A\,n} = m_{A\,n-1} + (e_{n-\frac{1}{2}} - f_{O\,n})\Delta t - (m_{L\,n} - m_{L\,n-1}) \tag{B10}$$

Here, $e_{n-\frac{1}{2}}$ is assumed to be known (though this only applies to the "forward" solution for atmospheric $CO_2$ from emissions, solving for emissions from $CO_2$ is also implemented in the model code).

After calculating the "committed" values $m_L^{c\Delta}{}_n$, $m_S^{c0}{}_n$ from the model state at $t_{n-1}$, equations (B7) through (B10) are solved

$$\Delta f_{\text{NPP}} = \frac{\frac{df_{\text{NPP}}}{dm_A}\big|_{n-1}}{UV + W}\bigg( m_{L\,n-1} - m_L^{c\Delta} + \Delta t\, e_{n-\frac{1}{2}} + \Delta t\, k_g A_O \Big(\varepsilon\, p_{S,n-1}^{CO_2} - m_{A\,n-1}$$

$$+ \varepsilon\, \frac{dp_S^{CO_2}}{dm_S}\bigg|_{n-1} \Big[m_S^{c0} - m_{S\,n-1} + \sum_k p_{fk\,O}\big(\frac{m_{L\,n-1} - m_L^{c\Delta}}{\Delta t} + e_{n-\frac{1}{2}}\big)\Big]\Big)\bigg) \tag{B11}$$

with the auxiliary variables

$$U = k_g A_O \, \varepsilon \left.\frac{\mathrm{d} p_S^{\mathrm{CO_2}}}{\mathrm{d} m_S}\right|_{n-1} \sum_k p_{fk_O} + 1 \tag{B12}$$

$$V = \left.\frac{\mathrm{d} f_{\mathrm{NPP}}}{\mathrm{d} m_A}\right|_{n-1} \sum_k p_{fk_L} + 1 \tag{B13}$$

$$W = \Delta t \, k_g A_O \tag{B14}$$

and, after inserting into equation (B6),

$$f_{O\,n} = \frac{k_g A_O}{U + W}\left(m_{A\,n-1} - \varepsilon \, p_{S,n-1}^{\mathrm{CO_2}} - \varepsilon \left.\frac{\mathrm{d} p_S^{\mathrm{CO_2}}}{\mathrm{d} m_S}\right|_{n-1} (m_S^{c0} - m_{S\,n-1}) - (m_{L\,n} - m_{L\,n-1}) + \Delta t \, e_{n-\frac{1}{2}}\right) \tag{B15}$$

The remaining variables are then calculated using equations (B7) and (B10), whereby first the components $m_{k\,n}$ are calculated as in equation (B4) and then summed. Finally, the nonlinear parametrizations are recalculated with the updated model state.

The order of these equations matters, as the updated variables are successively inserted into the following equations. The land part is solved first, and can be substituted by an explicit step or a separate model, while keeping the ocean step implicit.

An implicit time step is also implemented for calculating SAT from RF (again, solving RF from SAT is also implemented but not discussed here). $\mathrm{RF}(t_n)$ can be assumed as known, as atmospheric $CO_2$ is calculated first (i.e., no linearization necessary). Applying equation (B4) to temperature,

$$\Delta T_n c_S = \Delta T^{c\Delta} c_S + \Delta f_O^H \sum_k p_{fk_O}$$

$$\Delta T^{c\Delta} = \sum_k \Delta T_{k\,n-1} p_{mk_O} + f_{O\,n-1}^H / c_S \sum_k p_{fk_O} \tag{B16}$$

where $\Delta T^{c\Delta}$ is the "committed temperature" for constant heat flux to the ocean, and $\Delta f_O^H = f_{O\,n}^H - f_{O\,n-1}^H$ is the change in heat flux over one time step. Equations (8, 9, B16) are solved for $f_O^H$,

$$f_{O\,n}^H = \frac{\mathrm{RF}_n - \frac{\mathrm{RF}_{2\times}}{\Delta T_{2\times}}\Delta T^{c\Delta} + f_{H\,n-1}\sum_k p_{fk_O} \frac{\mathrm{RF}_{2\times}}{\Delta T_{2\times} c_S}}{\frac{\mathrm{RF}_{2\times}}{\Delta T_{2\times} c_S}\sum_k p_{fk_O} + a_O/A_O} \tag{B17}$$

Temperature change $\Delta T_n$ then follows from equation (B16).

The case of piecewise linear approximation (B3) differs from the piecewise constant one ( (B2)) only in a non-zero contribution of $f_{n-1}$ and a slightly different budget equation,

$$m_{A\,n} = m_{A\,n-1} + \left(e_{n-\frac{1}{2}} - \frac{f_{O\,n} + f_{O\,n-1}}{2}\right)\Delta t - (m_{L\,n} - m_{L\,n-1}) \tag{B18}$$

The first difference merely changes the calculation of "committed" changes, and only the second difference affects the solution
of the implicit time step. In practice, however, this can be neglected without loss of accuracy, and thus equations (B11 – B15) and (B17) are also used to solve the piecewise linear system (while equation (B18) is used to close the budget).

## B3 Temperature dependent parameters

BernSCM allows for temperature-dependent model parameters for IRF-based substitute models. This generalization of the IRF-approach is possible using a box-model form (section 2.3). Currently, temperature-dependent coefficients and time scales are implemented for the HRBM land biosphere substitute model (Appendix A2).

5       BernSCM updates any temperature-dependent model parameters by approximating the current temperature $\Delta T_n$ by the "committed" temperature $\Delta T^{c\Delta}$ as defined in equation (B16). Accuracy is further improved by substituting $\Delta T^{c\Delta}$ for $\Delta T_n$ in evaluating equation (B8) with temperature dependent parametrizations.

*Competing interests.*   The authors declare that they have no conflict of interest.

*Acknowledgements.*   This work received support by the Swiss National Science Foundation (#200020_172476).

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

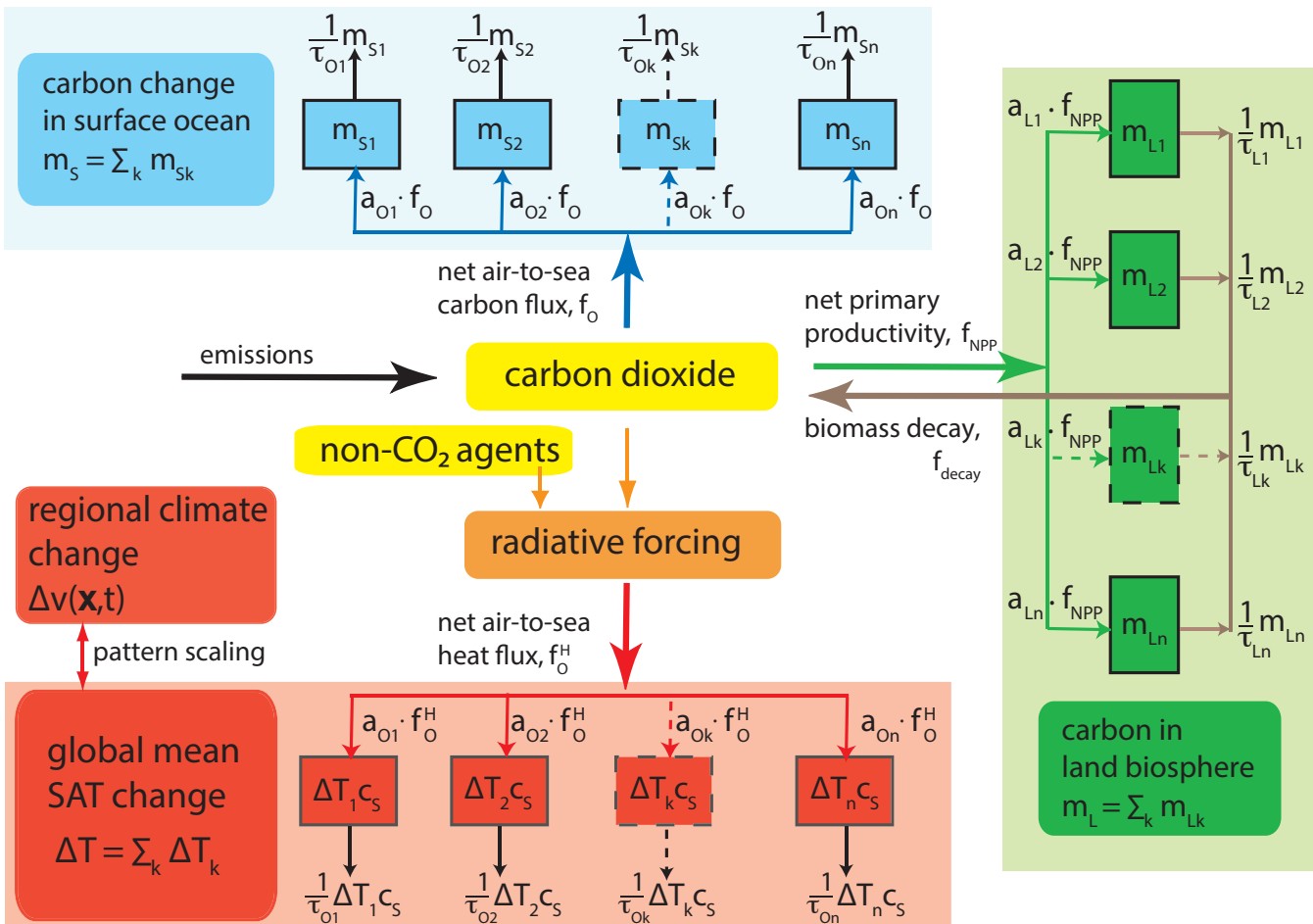

**Figure 1.** BernSCM as a box-type model of the carbon cycle-climate system based on impulse response functions. Heat and carbon taken up by the mixed ocean surface layer and the land biosphere, respectively, is allocated to a series of boxes with characteristic time scales for surface-to-deep ocean transport ($\tau_O$) and of terrestrial carbon overturning ($\tau_L$). The total perturbations in land and surface ocean carbon inventory and in surface temperature are the sums over the corresponding individual perturbations in each box, ($m_{Sk}, \Delta T_k, m_{Lk}$). Using pattern scaling, the response in SAT can be translated to regional climate change for fields $v(\mathbf{x}, t)$ of variables such as SAT or precipitation.

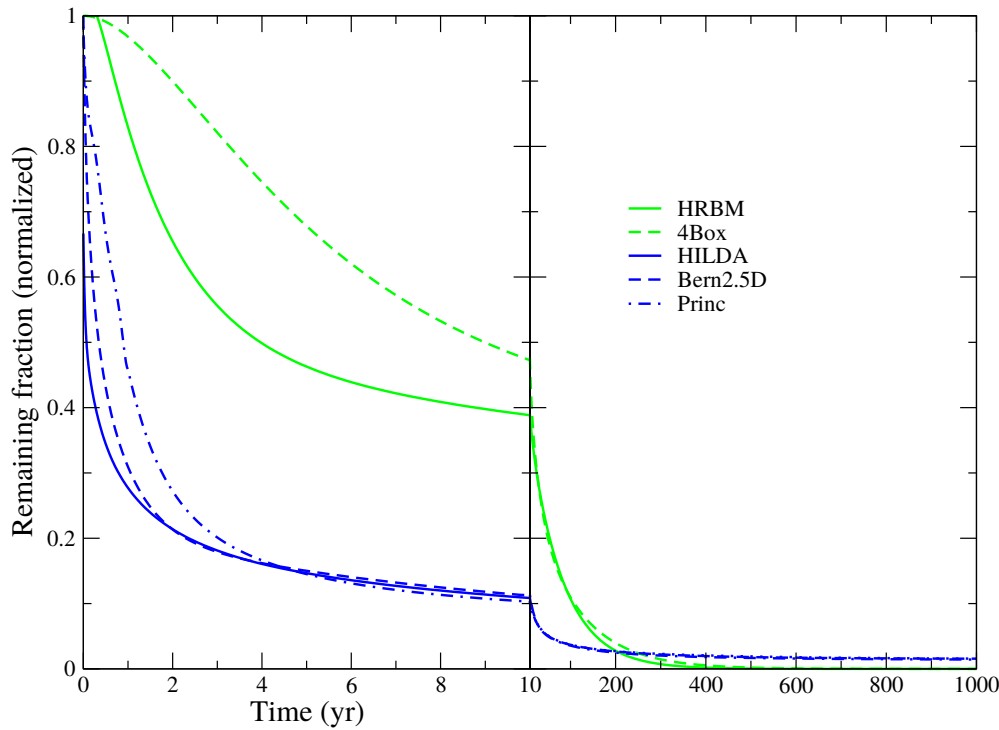

**Figure 2.** IRFs of ocean (blue) and land (green) model components (without temperature dependence). Ocean components are normalized to a common mixed layer depth of 50m (multiplied by $H_{\mathrm{mix}}/50$m), causing initial response to deviate from 1.

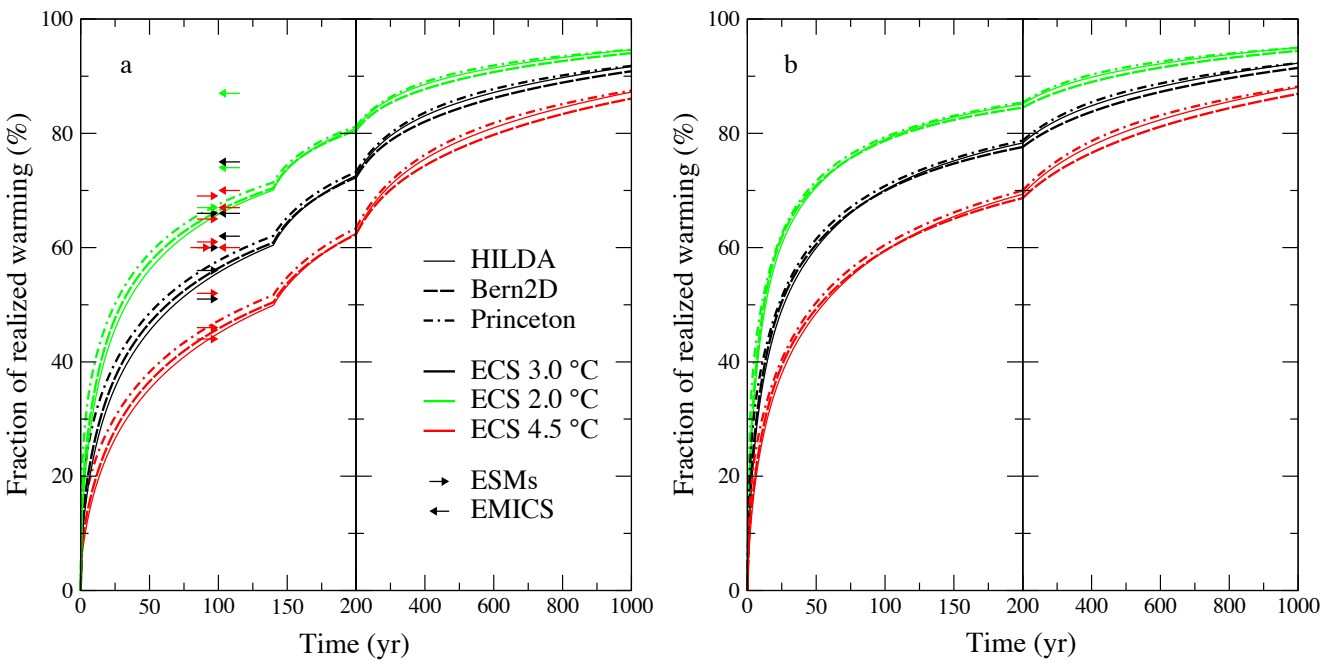

**Figure 3.** Fraction of realized warming (temperature divided by the equilibrium temperature for the current RF) for idealized experiments with prescribed atmospheric $CO_2$ concentration increase from preindustrial; panel a shows an exponential $CO_2$ increase by 1% per year over 140 years to approximately four times the preindustrial concentration (and linear increase in RF); panel b shows an abrupt increase to fourfold $CO_2$ concentration. BernSCM simulations are shown for climate sensitivities of 2, 3, and 4.5 K and the three available ocean model substitutes as indicated in the legend. Arrows in panel a indicate the corresponding warming fractions at year 99 compiled by (Frölicher and Paynter, 2015, SI Tables 1,2 ) for Earth System Models (ESM, right-pointing) and Earth System Models of Intermediate Complexity (EMICS, left-pointing); arrow colors indicate climate sensitivities below 2.5 K (green), between 2.5 and 3.5 K (black), and above 3.5 K (red).

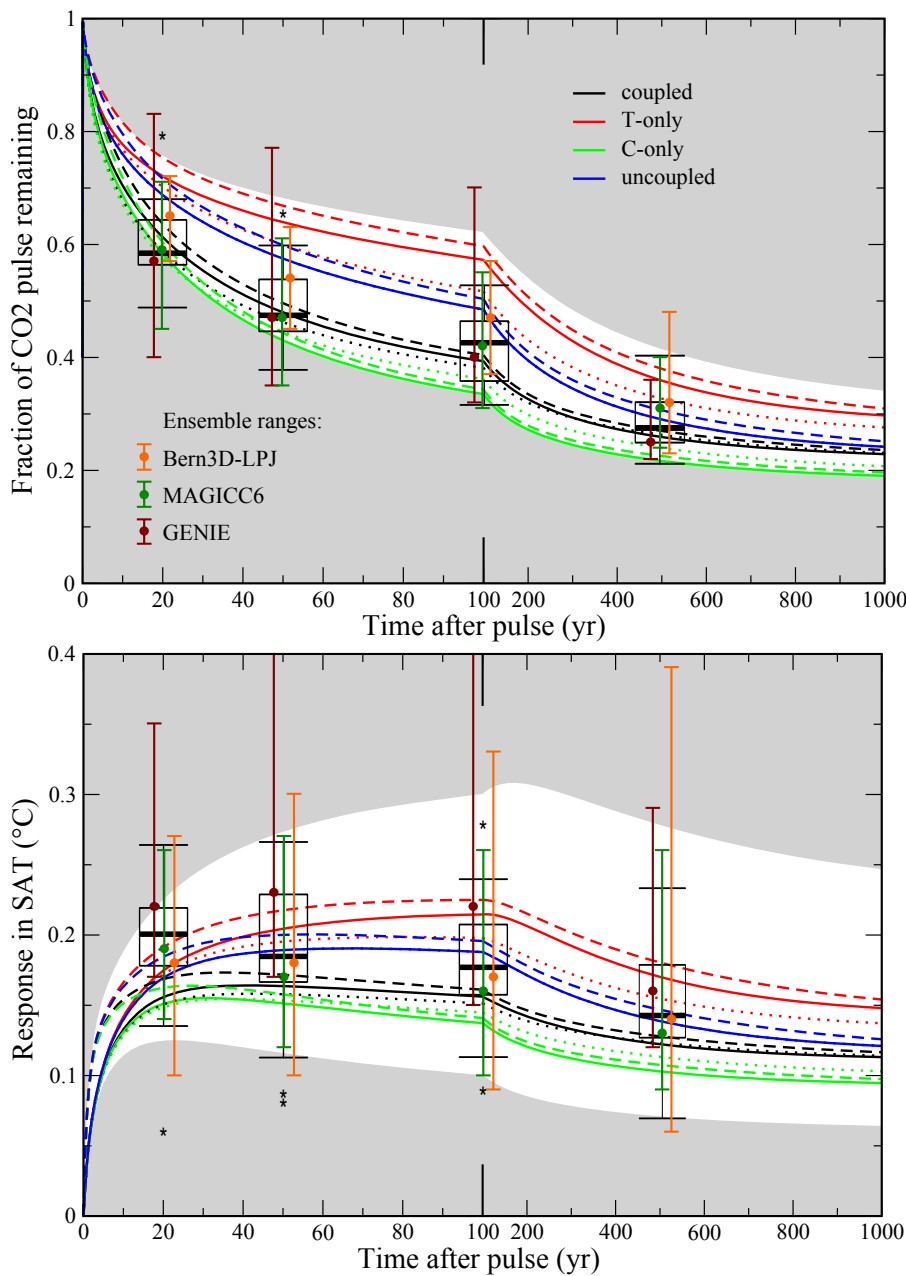

**Figure 4.** IRFMIP pulse response range compared to BernSCM range for parameter uncertainty (colors according to legend) and structural uncertainty, with model versions HILDA/HRBM (solid lines), HILDA/4box (dots), Princeton/HRBM (dashed). Standard climate sensitivity is 3°C, and a climate sensitivity range of 2–4.5°C is shown by the white area (envelope of all BernSCM runs). Single-model ensemble ranges from IRFMIP are included as errorbars indicating the 5-95% range and dots indicating the median. The multimodel IRFMIP range is shown by boxplots indicating median (bold black line), first quartiles (box), extreme values (whiskers) excluding outliers deviating from the median by more than 1.2 times the interquartile distance (asterisks).

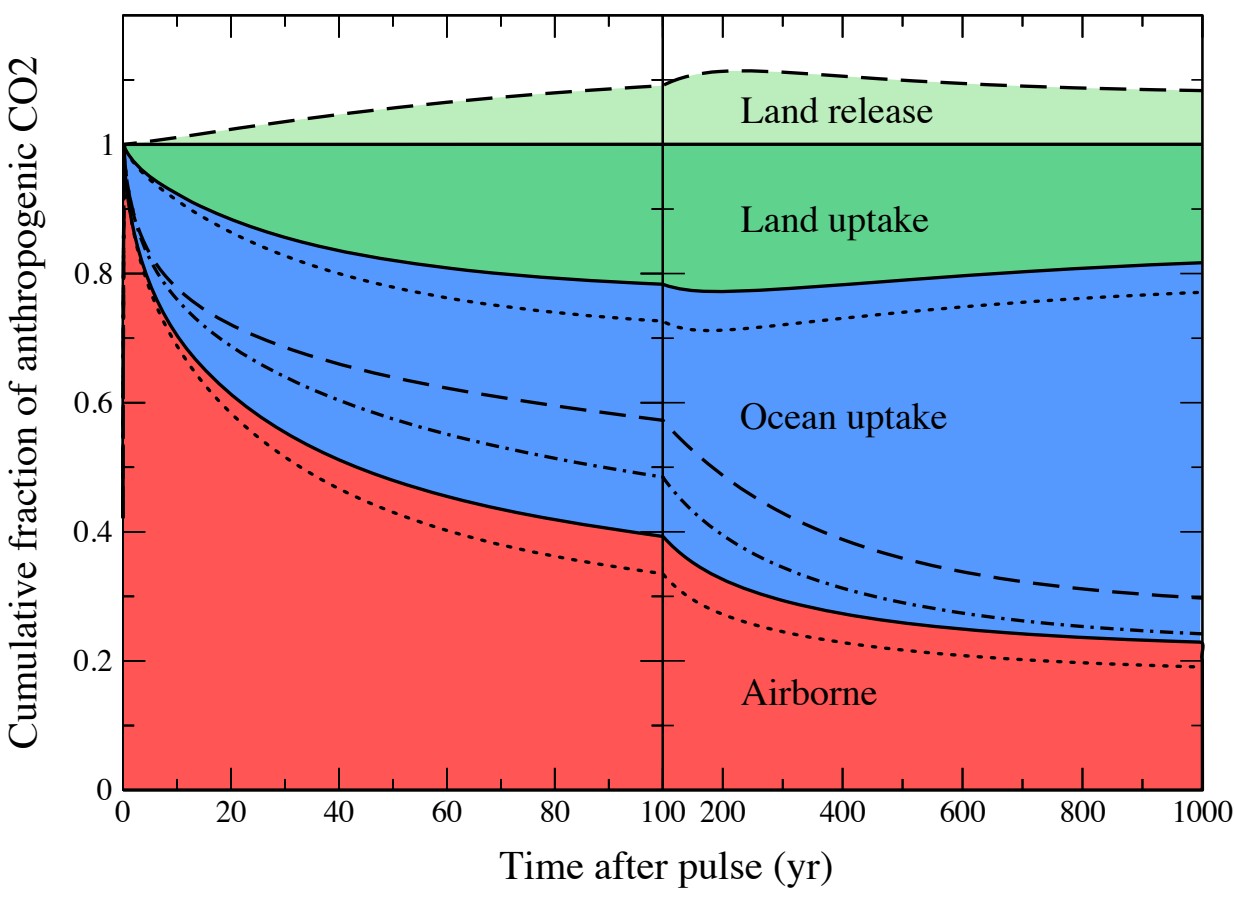

**Figure 5.** Land, ocean, and airborne fractions of the 100 GtC $CO_2$ pulse shown in Figure 4 for the coupled (solid lines and colored areas), the T-only (dashed), the C-only (dotted) and the uncoupled (dash-dotted) model setup. In the T-only case, the land biosphere exhibits a net release (light green shading), and the ocean uptake consists of the sum of this area and the area delimited by the dashed line below the line at 1; for the uncoupled case, land uptake is zero and ocean uptake extends from the dash-dotted line to unity.

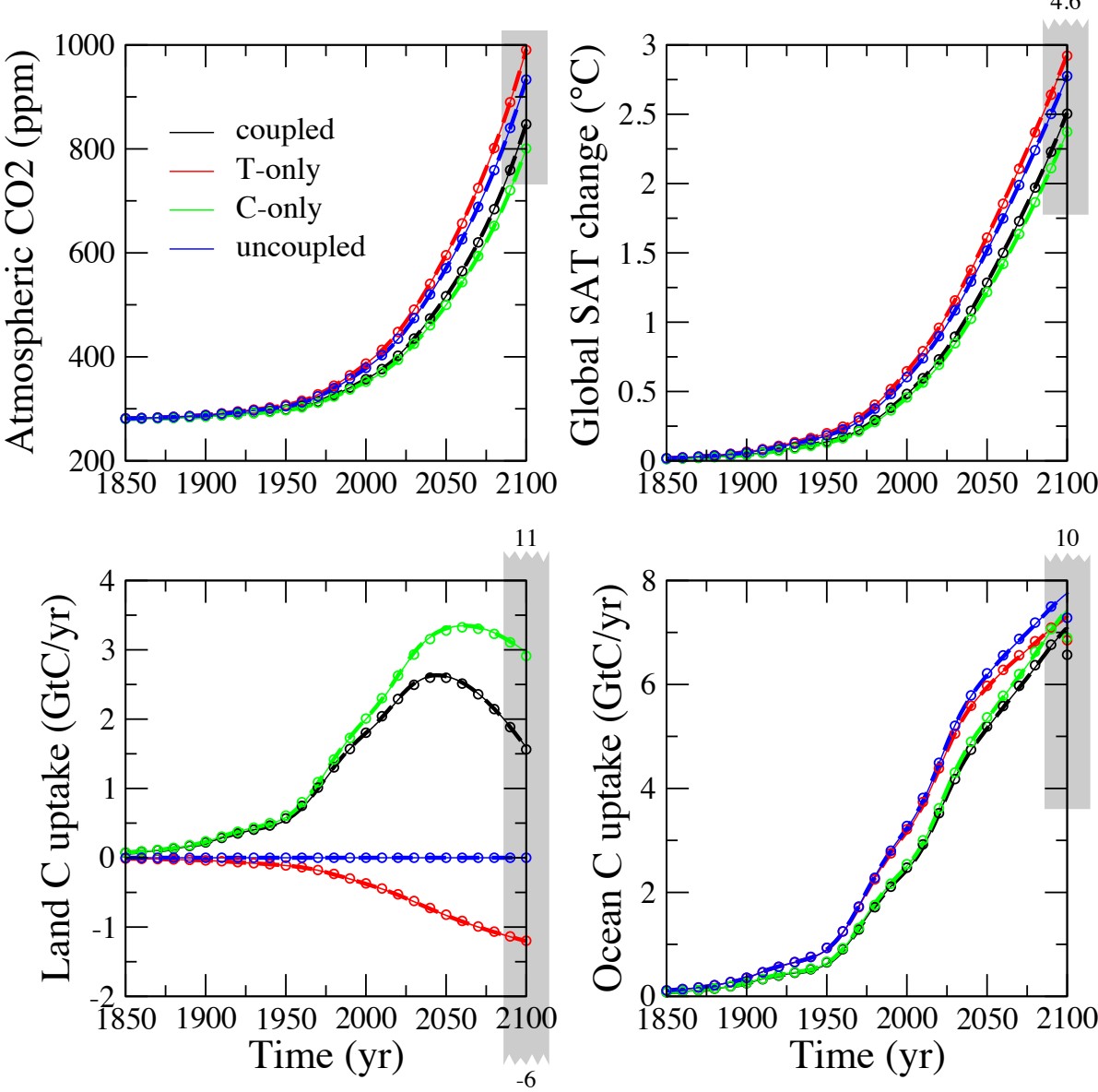

**Figure 6.** BernSCM simulations of the SRES A2 scenario used for C4MIP, with a climate sensitivity of 2.5°C and the HILDA/HRBM ocean/land components. Results for three numerical schemes are overlaid; 0.1 yr Euler forward timestep (solid thin line), ii. 1 yr implicit timestep (dashed bold line), iii. 10 yr implicit timestep with piecewise linear approximation of fluxes (circles); the difference at this resolution is only visible in the C uptake. The C4MIP model range at 2100 is indicated by grey bars; numbers above or below the bars indicate values outside of the chart range.

**Table 1.** C4MIP sensitivity metrics. The BernSCM range covers the carbon cycle settings as discussed in section 3.1, and different combinations of model components (HILDA-HRBM, HILDA-4box, Princeton-HRBM); the C4MIP range covers all participating models.

| | $\alpha$ | $\beta_L$ | $\beta_O$ | $\gamma_L$ | $\gamma_O$ | $g$ |
|---|---|---|---|---|---|---|
| Unit | $10^{-3}\,\frac{^\circ C}{\mathrm{ppm}}$ | $\frac{\mathrm{GtC}}{\mathrm{ppm}}$ | $\frac{\mathrm{GtC}}{\mathrm{ppm}}$ | $\frac{\mathrm{GtC}}{\mathrm{K}}$ | $\frac{\mathrm{GtC}}{\mathrm{K}}$ | $10^{-2}$ |
| | | | BernSCM | | | |
| Standard | 4.4 | 0.75 | 1.2 | -46 | -31 | 8.3 |
| Range | 4.1–4.6 | **0**–0.75 | 1.0–1.2 | -46–**0** | -31–**0** | 0–8.4 |
| | | | C4MIP ensemble | | | |
| Average | 6.1 | 1.35 | 1.13 | -79 | -30 | 15 |
| Range | 3.8–8.2 | 0.2–2.8 | 0.8–1.6 | -177– -20 | -67– -14 | 4–31 |

**Table A1.** Model variables

| Variable | Meaning | Unit |
|---|---|---|
| $m_A$ | Atmospheric $CO_2$ carbon | GtC |
| $m_L$ | Land biomass carbon | GtC |
| $m_S$ | Dissolved inorganic C perturbation in ocean mixed layer | GtC |
| $\Delta\text{DIC}$ | Perturbation of dissolved inorganic C concentration in mixed layer | $\mu$mol/kg |
| $p_{A/S}^{CO_2}$ | Atmospheric/ocean surface $CO_2$ pressure | ppm |
| $RF$ | Radiative forcing | $\text{Wm}^{-2}$ |
| $\Delta T$ | Global mean surface (ocean) temperature perturbation | °C |
| $\Delta T^{eq}$ | Equilibrium $\Delta T$ for current RF | °C |
| $e$ | $CO_2$ emissions | GtC/yr |
| $f_O$ | Air-sea C flux | GtC/yr |
| $f_{\text{deep}}$ | Net C flux from mixed layer to the deep ocean | GtC/yr |
| $f_{\text{NPP}}$ | NPP | GtC/yr |
| $f_{\text{decay}}$ | Decay of terrestrial biomass C | GtC/yr |
| $f_O^H$ | Air-sea heat flux | W |
| $f_{O\ \text{deep}}^H$ | Net heat flux from mixed layer to the deep ocean | W |

**Table A2.** Model parameters

| Parameter | Meaning | Unit | HILDA | Bern2D | Princeton |
|---|---|---|---|---|---|
| $H_{\mathrm{mix}}$ | Depth of mixed ocean surface layer | m | 75 | 50 | 50.9 |
| $A_O$ | Ocean surface area | $\mathrm{m}^{-2}$ | $3.62{\cdot}10^{14}$ | $3.5375{\cdot}10^{14}$ | $3.55{\cdot}10^{14}$ |
| $k_g$ | Gas exchange coefficient | $\mathrm{yr}^{-1}A_O^{-1}$ | 1/9.06 | 1/7.46 | 1/7.66 |
| $T^*$ | Global average ocean surface temperature | °C | 18.17 | 18.30 | 17.70 |
| | | | All models | | |
| $a_O$ | Ocean fraction of earth surface | - | 0.71 | | |
| $\varepsilon$ | Atmospheric mass of C per mixing ratio | GtC/ppm | 2.123 | | |
| $\varrho$ | Density of ocean water[a] | $\mathrm{kg/m}^3$ | 1028 (1026.5) | | |
| $c_p$ | Specific heat capacity of water | J/kg/K | 4000 | | |
| $c_s$ | Mixed layer heat capacity | J/K | $c_p\,\varrho H_{\mathrm{mix}}A_O$ | | |
| $M_{\mu\mathrm{mol}}$ | Mass of DIC per micromole | gC/$\mu$mol | $12.0107\cdot10^{-6}$ | | |
| $\mathrm{RF}_{2\times}$ | RF per doubling of atm. $CO_2$ | $\mathrm{Wm}^{-2}$ | 3.708 | | |
| $\Delta T_{2\times}$ | Equilibrium climate sensitivity for $CO_2$ doubling | °C | free | | |

[a]The first value is used in the climate component equations, the value in parentheses in the C cycle component equations.

**Table A3.** Mixed-layer IRF/Box parameters

| | | | | | | | | | |
|---|---|---|---|---|---|---|---|---|---|
| **HILDA** | | | | | | | | | |
| Input coefficients | $a$ | (-) | 0.27830 | 0.24014 | 0.23337 | 0.13733 | 0.051541 | 0.035033 | .022936 |
| Time scales | $\tau$ | (yr) | 0.45254 | 0.03855 | 2.1990 | 12.038 | 59.584 | 237.31 | |
| **Bern2.5D** | | | | | | | | | |
| Input coefficients | $a$ | (-) | 0.27022 | 0.45937 | 0.094671 | 0.10292 | 0.0392835 | 0.012986 | .013691 |
| Time scales | $\tau$ | (yr) | 0.07027 | 0.57621 | 2.6900 | 13.617 | 86.797 | 337.30 | |
| **Princeton GCM** | | | | | | | | | |
| Input coefficients | $a$ | (-) | 2.2745 | -2.7093 | 1.2817 | 0.061618 | 0.037265 | 0.019565 | 0.014818 |
| Time scales | $\tau$ | (yr) | 1.1976 | 1.5521 | 2.0090 | 16.676 | 65.102 | 347.58 | |

**Table A4.** Land C stock IRF/Box parameters

| HRBM | | | | | | | |
|---|---|---|---|---|---|---|---|
| Input coefficients | $a$ | (-) | -0.15432 | 0.56173 | 0.074870 | 0.41366 | 0.10406 |
| Time scales | $\tau$ | (yr) | 0.20107 | 1.4754 | 8.8898 | 74.098 | 253.81 |
| sensitivities | $s_a$ | (-) | 0.14 | 0.056 | 0.072 | 0.044 | 0.069 |
| | $s_\tau$ | (-) | 0.056 | 0.079 | 0.057 | 0.053 | 0.036 |
| **4Box** | | | | | | | |
| Input coefficients | $a$ | (-) | -1.5675 | 2.0060 | 0.26828 | 0.29323 | |
| Time scales | $\tau$ | (yr) | 2.1818 | 2.8571 | 20 | 100 | |

**Table A5.** Performance and accuracy for time steps 1–10 yr relative to a reference with a time step of 0.1 yr. The reference simulation is solved explicitly, otherwise an implicit solution was used. The average execution time of the time integration loop is given as a fraction of the explicit case. For atmospheric $CO_2$ and SAT, the root mean square difference to the explicit case, divided by the value range over the simulation is given. All values are for the C4MIP A2 scenario (years 1700 – 2100), using the HILDA ocean component and the HRBM land component with standard temperature and carbon cycle sensitivities (coupled).

| $\Delta t$ | 1yr | 10yr |
|---|---|---|
| discretization | piecewise const. | piecewise lin. |
| execution time | 15% | 2 % |
| CO2 RMS/range | 0.31‰ | 0.45‰ |
| SAT RMS/range | 0.52‰ | 0.53‰ |