# Peer review of "The Bern Simple Climate Model (BernSCM) v1.0: an extensible and fully documented open source reimplementation of the Bern reduced form model for global carbon cycle-climate simulations"

_Geoscientific Model Development, 2017_

## Referee Comment (RC1) · H. Metzler (Referee) · 1 Dec 2017

Report on

**Kuno Strassmann and Fortunat Joos**

**The Bern Simple Climate Model (BernSCM) v1.0: an extensible and fully documented open source reimplementation of the Bern reduced form model for global carbon cycle-climate simulations**

Holger Metzler

December 1, 2017

**General comments**

This very interesting paper describes the Bern Simple Climate Model (BernSCM) v1.0. BernSCM simulates relations between $CO_2$ emissions, atmospheric $CO_2$, radiative forcing (RF), global mean surface air temperature (SAT), as well as carbon and heat

fluxes between atmosphere, ocean, and land biosphere. It is a reduced form coupled carbon-climate model that emulates more complex coupled models by replacing complicated components with nearly linear behaviour by impulse response functions (IRFs). This (to the best of my knowledge, novel) approach leads to a coupled carbon-climate model which is easy to understand and needs only low computational cost to be run. Comparisons with results from two multi-model intercomparison studies (C$^4$MIP; IRFMIP from Joos et al., 2013) show that BernSCM simulations give representative results with respect to current knowledge about carbon-climate interactions.

I am convinced that this manuscript can be scientifically important in two ways: 1) The practical application of the model itself or extended versions in its own right or as part of bigger models can lead to advances in multiple directions. 2) The theoretical foundations of the manuscript based on IRFs provide an interesting perspective on the theory of ecological modelling. Very appealing is the interpretation of the IRFs as representing parallel systems with multiple boxes, for example. Apart from some minor exceptions, the manuscript and in particular the appendix and the provided Fortran code of the model are carefully prepared. The authors took care that the BernSCM model and its implementation can be reproduced. Furthermore, the manuscript is well organized and the results are nicely presented.

There are some technical problems with the equations that describe the model and I suspect an inherent theoretical problem as soon as the IRFs become time-dependent or depend on other states of the system (e.g., temperature, $CO_2$). While the technical problems can be solved easily, I am not sure about the theoretical issue, as I will explain in more detail below.

Even if the theoretical issues cannot be completely resolved, I consider this manuscript worth for publication in Geoscientific Model Development, if the authors make the readers aware of the situation.

**Specific comments**

General explanation

The theoretical idea of this manuscript is very appealing. The IRFs used to substitute complex model components are provided by earlier simulations of highly complex models and just plugged into BernSCM. This makes the model structure pretty simple and the model can be used to understand ongoing processes on a global level without getting lost in distracting details, for example. Furthermore, the computation is very fast due to the use of the IRFs. This speed is even improved by disassembling the IRFs into their most important time scales, which allows an interpretation of the substituted IRFs as describing an underlying parallel multi-box model. This approach allows a very fast recursive computation which is carefully explained in great detail in the appendix and implemented in the provided Fortran package of BernSCM v1.0.

The unit issue

In some equations the units do not fit. The main reason is that $\varepsilon$ has been given the wrong explanation and the wrong unit in Table 2. The correct unit is GtC/ppm and a better description could be "mass of C per atmospheric concentration". This solves the unit problem in equations (5) and (25). Equation (8) should then be

$$p_A^{CO_2} = m_A \cdot \varepsilon^{-1}. \tag{1}$$

As far as I could see from the code, it is implemented correctly.

In equation (7) the units do not give the desired (Table 1) $\mu$mol/kg. To that end the unit of $M_{\mu mol}$ needs to be changed to gC/$\mu$mol.

Very confusing is also the use of different time units like in equation (9). Carbon fluxes are measured per year and heat fluxes per second (W=J/s). Nevertheless the integral limits are in both cases $t_0$ and $t$. I could not find any correction term in the manuscript. In the code this correction seems to be made.

In equation (10), the unit results in W, not in PetaW as stated in Table 1. Also here a correction term is necessary. Again, this seems not to be an issue in the code.

Linear equations and IRFs

Equation (14) is only true if $m(t_0) = 0$. The general equation for the state $m$ at time $t$ is

$$m(t) = r(t - t_0)\, m(t_0) + \int_{t_0}^{t} f(t')\, r(t - t')\, dt'. \tag{2}$$

Since the authors use equation (14) to compute perturbations with an equilibrium value $m(t_0) = 0$, this does not lead to problems, but the way equation (14) is described is mathematically not correct. I have the feeling this happened, because the authors from the beginning had a perturbation with equilibrium equal to zero in mind, but started the section then with a slightly more general set up. Line 17 on page 5 does not mention perturbations.

The infinite time scale issue

When equation (20) is inserted in (14) to obtain equation (21), $a_\infty$ somehow disappears. As soon as $a_\infty \neq 0$ (ocean IRF), a term is missing in equation (21). The

equation should then look like

$$m(t) = \int_{t_0}^{t} f(t')\,a_\infty\,dt' + \sum_k \int_{t_0}^{t} f(t')a_k\,e^{-(t-t')/\tau_k}\,dt'. \tag{3}$$

If $f(t') \geq c > 0$, then the first part of the equation goes to infinity as $t \to \infty$ and the perturbation grows indefinitely. If some constant share from a constant input is never going to be decayed, this share accumulates forever. Also the carbon coming from carbon conversion in the ocean model (page 7, line 20), is going to decay at some point. A constant share of remaining carbon should not result from a multiplication with an input flux coming from the atmosphere. The same explosion effect can be seen in equation (A1). Also in Equation (A14) the to $a_\infty$ associated term $\frac{1}{2}\,a_\infty\,(\Delta t)^2$ is missing. It cannot already be included in the present sum, because $B_\infty = \frac{1}{2}\,a_\infty$ is never going to be multiplied with an exponential.

I do not know how this problem is handled in the implementation of the model.

The theoretical issue

In my opinion, the theoretical foundations of this model are sound as long as the substitute IRFs are time-independent and also independent of other state variables. However, the great power of BernSCM emerges when temperature or $CO_2$ dependencies are explicitly allowed in the IRFs. I am not perfectly sure, if the theory behind the IRF approach is still valid in this case, even though the simulations show reasonable results.

I think it is important to stress the fact, that equation (14) works for time-dependent forcings, but for time-independent processes only. The impulse response function $r$ here depends only on the difference $t - t'$ of the time $t$ at which we are interested in the perturbation $m(t)$ and the time $t'$ at which the input $f(t')$ came into the system.
The absolute time $t'$ is not used by the impulse response function $r$. Consequently, the underlying black-boxed process which can be modelled by this approach is assumed to be time-independent (has constant coefficients). If now the IRF depends additionally on temperature, the impulse response function needs to "know" the current time $t'$ and becomes $\tilde{r}(t', t - t')$. A different forcing function $f$ leads to a different system state which then results in a different IRF. The system is inherently non-linear, even though it looks linear. From the code (and unfortunately only from there) I could see that the IRFs are provided as a set of coefficients $a_k$ and a set of time scales $\tau_k$. Probably these numbers result from an analysis of complex simulations (e.g. HRBM). If so, the numbers come from a non-linear model and this very IRF is representative only for this very model run. In a non-linear setting, a different model run (initial value, temperature, sensitivities) could theoretically lead to a very different IRF which is then going to be ported to BernSCM. The analysis in section 4 shows that this does not have drastic influence here, probably in part because the external emission forcing was chosen to be the same (SRES A2).

Additional to this possible dilemma, the IRF comes with additional numbers for temperature sensitivity. These numbers are used in each time step to adapt the IRF in dependence of current mean surface air temperature. As mentioned above, $r(t - t')$ becomes $\tilde{r}(t', t-t')$. In the derivation of equation (A7), which is crucial for the numerical implementation, this leads to a problem. The term

$$R_i = 1/\Delta t \int\limits_{t_{i-1}}^{t_i} r(x)\,dx \qquad (4)$$

becomes

$$\tilde{R}_i = 1/\Delta t \int\limits_{t_{i-1}}^{t_i} \tilde{r}(t - x, x)\,dx \qquad (5)$$

and the integration becomes much more difficult, in particular if both $a_k$ and $\tau_k$ are temperature dependent.

I am not sure whether this problem can be discussed away by numerical means or even by purely theoretical considerations, but at the moment I have a strange gut feeling about this issue.

**Technical corrections**

- in general: Punctuation around equations is missing very often, in particular full stops or commas after the equations when necessary.

- in general: Some abbreviations are never introduced, e.g., HRBM, HILDA. Sometimes the explanation of the abbreviation comes late in the text. This happens in particular when reading the figure captions and figures are referred to in more places. Maybe this is hard to circumvent without destroying the text flow.

- There is a mix of British (analyse) and American ("behavior") English.

- in general: It is difficult to find out which constant means what since tables 1 and 2 are not complete. Some terms are explained in the text, some in the tables.

- page 1, line 14: "in an spatially"

- page 2, line 34: "of BernSCM a an IAM component"

- page 2, line 35: "managment"

- page 3, line 1: "cycle assessments(Levasseur et al., 2016)":
  space before parenthesis

- page 3, line 26: What does "LULUC" stand for?

- page 4, line 4: "by" or colon missing at the end?

- page 4, line 7, equation (4): How is $\varphi_{NPP}$ defined?

- page 4, line 11: "Ao" instead of $A_O$, "eps" is probably meant to be "$\varepsilon$"

- page 4, line 14, equation 8: How are the functions $\psi$ and $\chi$ defined?

- page 5, line 1: "and the separation of SAT from radiative equilibrium":
  Does "separation" here refer to the difference between $1$ and the ratio $\frac{\Delta T}{\Delta T^{eq}}$? To me the word "separation" is rather confusing in this context.

- page 5, line 9, equation (12): I could not find a description of $p_{A0}^{CO_2}$ anywhere. Typo?

- page 7, line 5, equation (21): The lower limit of the integral should be $t_0$ instead of $0$.

- page 7, line 15: "$a_{O_k}$" is called $a_k$ in Figure 1 (blue box). Also in the red box the constants are called $a_k$. Only in the green box they are called $a_{Lk}$. Similar problems with $\tau$.

- page 7, line 20: Which model from Table 3 is here referred to? The Bern2.5D or the 4-box Siegenthaler and Joos?

- page 8, lines 9-10: "here, the IRF substitutes for the HILDA ocean model, and the HRBM land biosphere model are used for the standard setup":
  It took me a while to understand this phrase. Maybe an additional "for" in front of "the HRBM" and omitting the comma are helpful?

- page 8, lines 13-14: "and the dependency of land C on temperature ($f_{decay}$) increases with warming, eq. (2))":
  From equation (2) I cannot see what happens with warming. Going to equation (19), I can see that it depends very much on $f_{NPP}$. This is defined in equation (4) and depends heavily on $\varphi_{NPP}$, which is not explained at all.

- page 9, line 4, equation 25: An interchange of $\beta_O$ and $\beta_L$ on the right hand side makes it better comparable with the left hand side and the following text.

- page 9, line 7-8: "$\beta$ is the change in carbon stored (in GtC)":
  Following Table 3, $\beta$ has the unit GtC/ppm and in Friedlingstein et al. (2006) it is referred to as "sensitivity of land carbon storage to atmospheric $CO_2$". If I understood equation (25) correctly, this is a more precise and less confusing description. The same holds for $\gamma$.

- page 9, line 26: I did not immediately recognise "airborne fraction" as a technical term. Maybe a short explanation could avoid confusing non expert readers.

- page 10, line 1: "(Figure. 3)"

- page 10, line 1-2: Why different units?

- page 10, lines 3-5: Looking at Figure 3 (upper panel), after 100 years all simulated values are greater than 0.3. Where does the value 0.3 from the text come from?

- page 10, lines 5-7: "For AF simulated with BernSCM, the standard coupled setup is close to the IRFMIP multimodel median, but the BernSCM uncertainty range is asymmetric. The IRFMIP multi-model range is similarly asymmetric.:"
  The word "but" confuses me, because the "IRFMIP multi-model range is similarly asymmetric".

- page 12, line 21: "structural simplicitly"

- page 13, line 15, equation (A4): The integral limits are interchanged. They do not change with the parameter transformation, because $dt' = -dx$ makes for a second sign change.

- page 15, line 2, equation (A14):
  Maybe it is better to write $(\Delta t)^2$.

- page 15, line 13: "explicite"

- page 15, line 20: "Equations (A1,A2)"
  Space between A1 and A2?

- page 16, line 13, equation (A20):
  Maybe $(m_{S_n} - m_{S_{n-1}})$ is correct? I am not sure.

- page 17, line 12: "explicite"

- page 17, line 19: "Equations (11,10,A28)"
  Space between equation numbers?

- page 17, line 24, equation (A30): Is it correct that $f_{O_n}$ appears twice in this formula?

- Figure 4: For me it is impossible to differentiate between dashed and dashed-dotted lines here. Maybe a different colour/line-style scheme could help here. Since the differences resulting from the use of different numerical schemes are almost invisible anyway, one could even go without trying to make them visible and simply mention that the differences are small. On the other hand, the point the authors want to emphasise here, is that due to the very small differences, the fastest scheme can be implemented. This leads to the entire appendix and the Fortran implementation. So it is rather an important point.

- Table 1: "$f_A$: net flux to atmosphere flux"

- Table 1: "$f_{deep}$: Flux mixed layer to deep":
  Why does "Flux" start with capital F? Missing "ocean" at the end?

- Table 2: Capitalization of first word in second column inconsistent?

- Table 2: From the units I think $c_p$ should be called "specific heat capacity".

- Code: Why is in the file parLandHRBM.inc the first weight negative? If I understood correctly, those weights are the $a_{L_k}$ values which here nicely sum to one, but how do you distribute a negative share of incoming carbon?

---

## Referee Comment (RC2) · Anonymous Referee #2 · 4 Dec 2017

This paper by Strassmann and Joos presents the reimplementation of the Bern Simple Climate Model (BernSCM), a reduced form model of the anthropogenic perturbation of the carbon-climate system. This is a historic model for the community, since it and its offspring have been used since the IPCC SAR. This new implementation is useful for the community, especially as this paper focuses on transparency and the model's code is provided in an open-source format.

Being an old model, the BernSCM ignores some relatively recent developments in climate sciences and modeling. In itself, it is not so much of a problem, as the authors leave the door open to further development of the model, both in the manuscript and in the model's code. However, mention and discussion of these caveats is required, especially regarding some specific points I develop below.

I also believe that the paper could benefit from a more careful rewriting, especially for some sections that I had to read several times – and I am still not 100% sure of what is done in some parts of the paper! In all honesty, some parts give the impression that the authors were in a rush for writing the paper.

So I fear publication can only be recommended if the few scientific issues I raise below are answered/discussed, and if the text itself is improved.

**Major points:**

**1.** My first point concerns the use of the same IRF parameters for the ocean carbon cycle and the climate system. If I understand it well, the function $r_O$ is the same for determining the ocean C sink and the temperature change, e.g. in equations (15) and (16). Although it would seem intuitive to use the same function, because – obviously – we are talking about the (same) world's ocean in both cases, I see several issues in doing so.

First, I am not quite sure one can assume that the diffusion process is the same for heat and for actual material such as carbon. (The assumption seems more reasonable for convection.) But more importantly, the biological pump does not affect heat transport, while it does for carbon. (Although, I am not sure whether there was a biological pump at all in the models used to calibrate the $r_O$ function – another thing worth being mentioned.)

Second, global patterns of heat uptake vs. carbon uptake are different. This means that one unit of incoming $f_O$ is dispatched differently than one of $f_O^H$, at the scale of

the global surface ocean. Therefore, it is likely that each of them is affected differently by the oceanic circulation. For the climate response, it is also known that this pattern affects an internal feedback (the ocean heat uptake feedback) in a way that changes the apparent time-scales of the climate response, see e.g. Geoffroy et al. (2013b) and references therein.

Third, the typical climate IRF only has two time-scales (e.g. Geoffroy et al., 2013a), and these are quite different from the time-scales from Joos et al. (1996). And, maybe more importantly, the typical two-box climate model implied by the typical climate IRF (Geoffroy et al., 2013a) includes a bidirectional exchange of energy between the surface and deep oceans. This is not the case in the assumed formulation presented here. There are some fundamental reasons for not having this bidirectional exchange for the carbon cycle: the so-called 'ocean invasion' is a slow process, and ultimately there is a sink of C in the deep ocean that involves geological chemical reactions (and time-scales). But can this be also applied to the climate system and heat transport?

Therefore, I believe this is an assumption made by the authors that $r_O$ can be applied to the climate system as well. Despite all of the above, it may still be acceptable. But it should be presented as such, and it also warrants a discussion in the text. Additionally, the response to a step of radiative forcing (typically 4x CO2) of this climate model has to be compared to that of more complex models. I strongly suggest adding a (sub)figure in which the BernSCM climate response is compared to that of CMIP5 models, taken e.g. from Geoffroy et al. (2013b). This would complement figure 3.

**2.** I was very troubled by section 3 and how the carbon-climate feedbacks are represented/investigated with BernSCM, in relation with C4MIP. At first, I thought BernSCM was trying to emulate the C4MIP models' sensitivities (which would have been a new feature).

In the end, my understanding is that the uncertainty range provided e.g. in table 4 is obtained by combining variations of: (i) the ocean model, 2 options; (ii) the land model,

2 options; (iii) the experimental setup, i.e. coupled/uncoupled/Tonly/Conly, 4 options. That is a total of 2×2×4=16 configurations. But my concern, here, is that I think that turning a process on or off can hardly be considered a new configuration of the model. Therefore, although the results shown e.g. in figure 3 or 4 are interesting, the ranges provided in table 4 are artificial and misleading.

Interactive
comment

More generally speaking, the text should make it clear that there are not many parameterizations available for the model, and so it does not cover the full range of existing multi-model uncertainty (and therefore, it cannot be used in a probabilistic fashion). Again, it is not so much of a problem in itself, but this has to be made very clear.

**3.** I have some trouble with the way the solving of the differential system is presented, but more importantly I believe there is a mistake with how the temperature-dependent parameters are implemented.

I am not convinced by the lengthy demonstration of appendix A1. Equations (A2) and (A3) are the 'results' of this section, and I believe the following demonstration is not needed. Equations (A2) and (A3) can simply be obtained by using the 'exponential integrator' method to solving a first-order differential system. Although not everyone may know this method, it could be summed up in one or two equations (and appropriate references) rather than be re-demonstrated from scratch.

Equations (A2) and (A3) are simply obtained by 'reminding the reader' that the solution to the differential system:

$$\frac{\mathrm{d}m}{\mathrm{d}t} = -\frac{m(t)}{\tau} + a\,F(t) \tag{1}$$

can be discretized by multiplying by $\exp(\frac{\delta t}{\tau})$ and integrating between $t_n$ and $t_{n+1} = t_n + \delta t$:

$$m_{n+1} = \exp{-\frac{\delta t}{\tau}}m_n + \int_0^{\delta t}\exp(-\frac{\delta t - s}{\tau})\,a\,F(t_n + s)\,\mathrm{d}s \tag{2}$$

where $m_{n+1} = m(t_{n+1}) = m(t_n + \delta t)$, and $\delta t$ is the time step.

The above equation is exact, but can hardly be solved. It is usual to assume that $F$ is constant over the small time period of $\delta t$, which leads to the solution:

$$m_{n+1} = \exp(-\frac{\delta t}{\tau})\, m_n + \tau\,(1 - \exp(-\frac{\delta t}{\tau}))\, a\, F(\tilde{t}) \tag{3}$$

which is basically equation (A2) and (A3) combined. $\tilde{t}$ remains to be chosen, e.g. to be $t_n$ (forward method), $t_{n+1}$ (backward), or any other fancier method possible. When assuming $\delta t$ = 10 yr and a $F(t)$ is linear between $t_n$ and $t_{n+1}$, one immediately finds the $\delta t^2$ equations.

So far, no fundamental problem with the authors' equations and text. I just believe it could be written in a more efficient and straightforward way. But a problem arises when one assumes that the time-scale $\tau$ varies with time (through e.g. temperature) so that we have in fact $\tau = \tau_0 + \Delta\tau(t)$. The exponential integrator method can still be applied, albeit by using $\tau_0$ and not $\tau$ in the exponential function.

To do so, it is easier to rewrite the differential equation as:

$$\frac{\mathrm{d}m}{\mathrm{d}t} = -\frac{m(t)}{\tau_0 + \Delta\tau(t)} + a\, F(t) \tag{4}$$

$$= -\frac{m(t)}{\tau_0} + \frac{m(t)\,\Delta\tau(t)}{\tau_0\,(\tau_0 + \Delta\tau(t))} + a\, F(t) \tag{5}$$

which completely changes the exponential integrator form:

$$m_{n+1} = \exp(-\frac{\delta t}{\tau_0})\, m_n \quad + \quad \int_0^{\delta t} \exp\left(-\frac{\delta t - s}{\tau_0}\right) a\, F(t_n + s)\, \mathrm{d}s$$
$$+ \quad \int_0^{\delta t} \exp(-\frac{\delta t - s}{\tau_0})\, \frac{m(t_n + s)\,\Delta\tau(t_n + s)}{\tau_0\,(\tau_0 + \Delta\tau(t_n + s))}\, \mathrm{d}s \tag{6}$$

leading to:

$$m_{n+1} = \exp(-\frac{\delta t}{\tau_0})\, m_n + \tau\,(1 - \exp(-\frac{\delta t}{\tau_0}))\, a\, F(\tilde{t}) + (1 - \exp(-\frac{\delta t}{\tau_0}))\, m(\tilde{t})\, \frac{\Delta\tau(\tilde{t})}{\tau_0 + \Delta\tau(\tilde{t})} \tag{7}$$

The latter equation raises the issue that it is virtually impossible to use with a backward approach since $\Delta\tau(t_{n+1})$ is not known. But a bigger issue is that, if I understand it correctly, the authors do not use this equation nor any equivalent. I believe they simply apply the equation of the case with constant $\tau$ but with a value of $\tau$ that changes through time. That is, they use the following equation:

$$m_{n+1} = \exp(-\frac{\delta t}{\tau_0 + \Delta\tau(t)}) \, m_n + (\tau_0 + \Delta\tau(t)) \, (1 - \exp(-\frac{\delta t}{\tau_0 + \Delta\tau(t)})) \, a \, F(\tilde{t}) \quad (8)$$

instead of the one above.

Unless the authors can prove the difference between the two is negligible, I am afraid there is a fundamental mistake in the solving of the model.

**4.** I believe the model should be completely described in the paper. I mean: formulations for e.g. functions $p_S^{CO_2}$, $\psi$, $\chi$, as well as all the parameter values should be given. The model is relatively simple, and there are not that many parameters. Even if the values can be accessed in the code, the fact that this paper is a model description makes it necessary to be as exhaustive as possible.

**Minor points:**

p. 1 (sec. 1): SCMs have many more usages than what is given here. Generally speaking, I find that the citations of this paper are too self-centered. I think everyone acknowledges the importance of the original Joos et al. (1996) paper, but much has been done since then regarding IRFs.

p. 3, l.13: The "essentially linear behavior" is an assumption of the model.

p.3, l. 16: IRFs are indeed equivalent to box-models, albeit with constant parameters!

p.3, l.25: The non-inclusion of LULCC could be discussed a little.

p.6 l.10: It is probably better to give all the equations, even if very similar.

p.6 l.20: "conversation" => "conservation" (probably many typos I missed...)

p.6 l.22: I don't think it is $10^4$ or $10^5$ k̲yr. Unit is probably yr.

p.7 l.2: At this stage, it is very unclear whether the response based on HRBM is a usual linear IRF calibrated with climate-carbon feedbacks on, so that those are linearized within the IRF, or if the time-scales of the response are indeed interactively changed by temperature during the simulation. Note also that I don't think the name "IRF" can be given to a model with time-varying parameters. I believe an IRF is the integrated form of the differential equation, which can be obtained only when the parameters do not vary with time. When they do, there is no integrated form, and the model is just a box model.

p.7 l. 13: Similarly, I would question the fact that the equation s̲hows that IRF and box model are equivalent. I think they are per definition. The only difference being that one is the integrated form of the other.

p.7 l. 20: Can cite Li et al. (2009) who provide a nice discussion on the (over)interpretation of those parameters.

p.7 l.25: It is more than 'they can be viewed'. Per construction, IRFs show the exponential eigenmodes of the system they are calibrated upon. Raupach (2014) or Enting (2007) provide some insights on this.

p.8 (sec. 3): I really find this section difficult to apprehend. It would benefit from some re-organizing, e.g. with a subsection on the beta/gamma framework, and then one on what it gives when applied to BernSCM. This is also the part that made me wonder whether C4MIP models were emulated or simply used for comparison.

p. 8, l.16: Table 3 does not provide any parameter value. . .

p.8, l. 28-29: Please, name those simulations "T-only" and "C-only". The dash makes a lot of difference when reading the text that follows!

p.9, l.9: I don't think alpha is the "transient climate sensitivity" in the usual sense. Find another name.

p.9, l.10: Which original paper?

p.9, l.31: The "combinations" remain quite unclear.

p.10, l.1: Inconsistent temperature units (this is in the whole paper).

p.10, l.2: More important comment related to my first major points. The choice of a climate sensitivity does not affect the time-scales of the climate response. However, it is know that a higher climate sensitivity implies a slower climate system (e.g. Baker Roe, 2011).

p.10, l.9-10: The last bit of this sentence is very uninformative.

p.10, l.12: 3.2K

p.10, l.15-17: I believe the fundamental reasons exposed in my major point number 1 also explain a lot, here. Hence the need to compare the climate response alone, and not coupled to the carbon cycle as in figure 3.

p.10, l.23: Those sensitivities are not defined. . .

p.10, l.29-32: I don't see the point of those sentences. Yes, the obtained sensitivities are zero. But this is per construction, since the uncoupled cases are used to investigate the sensitivity. This relates to my major point 2.

p.11, l.4: I believe it is 0.5K, according to figure 4. Also these values are for a fixed climate sensitivity. So I wonder how informative they are.

p.11 (sec. 5): I don't find all the discussion about BernSCM/C4MIP very convincing, for the reason already exposed above.

p.12, l.1: Yes, but that requires building EOFs on more complex models. Mention and citations needed here.

p.12, l.3: Note that regarding precipitation (and likely cloud cover as well), we now know that the response is forcing dependent (e.g. Shine et al., 2015; and references

therein).

p.12, l.10: Yes. But simple models usable in a probabilistic fashion already exist out there.

p.12, l.23: GWPs and other metrics require inclusion of non-CO2 species. So I'm not sure the sentence here is relevant.

p. 12, l.26: I don't like the use of "fixed", here. It is e.g. not influenced by external factors such as climate change.

p. 13 (sec. A1): As I wrote in my major points 3, I believe this section could be more straightforward.

p. 13 (sec. A2): This section is awfully complicated! It makes me wonder about several things, and I could not find the answer... Couldn't a solver be used for the backward method? Is the backward method solved with an exact solution, or is the method proposed an approximation? Does it have to be that complicated?

Also, I find the equations extremely difficult to follow. There are four (!!) levels of notation: $U, V, W$ refer to $p_{fk}$ which refer to $A_k$ which refer to the original parameters $\tau_k$ and $a_k$. I am convinced this part could be written (and implemented in the code?) in a much simpler way

p.12 (sec. A3): Again, not completely clear how the climate-carbon feedback is implemented. See major point 3.

p.25 (fig. 3): A representation of the land and ocean fractions could be provided. Also, see major point 1: the climate response alone should be shown somewhere (be it within figure 3 or separately).

p.26 (fig. 4): Maybe show ranges from C4MIP?

p.29 (tab. 3): I don't find this table very informative. Parameter values and functional forms should be provided instead.

p.30 (tab. 4): Using the words "parameters" is one of the things that made me wonder whether C4MIP models were used as input to BernSCM or just to compare outputs. I would call that e.g. "metrics".

**References:**

Baker, M.B. and G.H. Roe. The Shape of Things to Come: Why Is Climate Change So Predictable? Journal of Climate 2009 22:17, 4574-4589

Enting, I.G. Laplace transform analysis of the carbon cycle, In Environmental Modelling Software, Volume 22, Issue 10, 2007, Pages 1488-1497, ISSN 1364-8152, https://doi.org/10.1016/j.envsoft.2006.06.018.

Geoffroy, O., Saint-Martin, D., Olivié, D. J. L., Voldoire, A., Bellon, G., and Tytéca, S.: Transient Climate Response in a Two-Layer Energy-Balance Model. Part I: Analytical Solution and Parameter Calibration Using CMIP5 AOGCM Experiments, J. Climate, 26, 1841–1857, 2013a.

Geoffroy, O., D. Saint-Martin, G. Bellon, A. Voldoire, D.J. Olivié, and S. Tytéca. Transient Climate Response in a Two-Layer Energy-Balance Model. Part II: Representation of the Efficacy of Deep-Ocean Heat Uptake and Validation for CMIP5 AOGCMs. Journal of Climate 2013b 26:6, 1859-1876

Joos, F., Bruno, M., Fink, R., and Siegenthaler, U.: An efficient and accurate representation of complex oceanic and biospheric models of anthropogenic carbon uptake, Tellus B, 48, 397–417, http://onlinelibrary.wiley.com/doi/10.1034/j.1600-0889.1996.t01-2-00006.x/abstract, 1996.

Li, S., Jarvis, A. J., and Leedal, D. T.: Are response function representations of the global carbon cycle ever interpretable?, Tellus B, 61, 361–371, 2009.

Raupach, M. R.: The exponential eigenmodes of the carbon-climate system, and their implications for ratios of responses to forcings, Earth Syst. Dynam., 4, 31-49, https://doi.org/10.5194/esd-4-31-2013, 2013.
[Figure]

Shine, K. P., Allan, R. P., Collins, W. J., and Fuglestvedt, J. S.: Metrics for linking emissions of gases and aerosols to global precipitation changes, Earth Syst. Dynam., 6, 525-540, https://doi.org/10.5194/esd-6-525-2015, 2015.

---

## Author Comment (AC1) · 2 Mar 2018

**Response to the comments by the reviewers.**

We thank both reviewers for their effort and commitment to review our manuscript in great detail. This is very much appreciated. We considered all comments and modified the manuscript in response. In particular, we extended the introduction and discussion to refer the reader to other simple climate models in the literature and to better explain the main features of BernSCM. Additional figures have been added as requested. The text flow in section 3 and 4 has been reorganized. The main conclusions and results remain unchanged compared to the previously submitted version.

For convenience, we repeat the comments by the reviewers below. The answers are given in indented text.

A revised version with the changes highlighted is attached to this response.

**H. Metzler (Referee)**

**General comments**

This very interesting paper describes the Bern Simple Climate Model (BernSCM) v1.0. BernSCM simulates relations between CO2 emissions, atmospheric CO2, radiative forcing (RF), global mean surface air temperature (SAT), as well as carbon and heat fluxes between atmosphere, ocean, and land biosphere. It is a reduced form coupled carbon-climate model that emulates more complex coupled models by replacing complicated components with nearly linear behavior by impulse response functions (IRFs). This (to the best of my knowledge, novel) approach leads to a coupled carbon climate model which is easy to understand and needs only low computational cost to be run. Comparisons with results from two multi-model intercomparison studies (C4MIP; IRFMIP from Joos et al., 2013) show that BernSCM simulations give representative results with respect to current knowledge about carbon-climate interactions. I am convinced that this manuscript can be scientifically important in two ways: 1) The practical application of the model itself or extended versions in its own right or as part of bigger models can lead to advances in multiple directions. 2) The theoretical foundations of the manuscript based on IRFs provide an interesting perspective on the theory of ecological modelling. Very appealing is the interpretation of the IRFs as representing parallel systems with multiple boxes, for example. Apart from some minor exceptions, the manuscript and in particular the appendix and the provided Fortran code of the model are carefully prepared. The authors took care that the BernSCM model and its implementation can be reproduced. Furthermore, the manuscript is well organized and the results are nicely presented.

> Thank you for these nice words

There are some technical problems with the equations that describe the model and I suspect an inherent theoretical problem as soon as the IRFs become time-dependent or depend on other states of the system (e.g., temperature, CO2). While the technical problems can be solved easily, I am not sure about the theoretical issue, as I will explain in more detail below.

Even if the theoretical issues cannot be completely resolved, I consider this manuscript worth for publication in Geoscientific Model Development, if the authors make the readers aware of the situation.

> Please see the answer to the specific comments below.

**Specific comments**

General explanation

The theoretical idea of this manuscript is very appealing. The IRFs used to substitute complex model components are provided by earlier simulations of highly complex models and just plugged into BernSCM. This makes the model structure pretty simple and the model can be used to understand ongoing processes on a global level without getting lost in distracting details, for example. Furthermore, the computation is very fast due to the use of the IRFs. This speed is even improved by disassembling the IRFs into their most important time scales, which allows an interpretation of the substituted IRFs as describing an underlying parallel multi-box model. This approach allows a very fast recursive computation which is carefully explained in great detail in the appendix and implemented in the provided Fortran package of BernSCM v1.0.

Thank you for these remarks.

The unit issue
In some equations the units do not fit. The main reason is that $\varepsilon$ has been given the wrong explanation and the wrong unit in Table 2. The correct unit is GtC/ppm and a better description could be "mass of C per atmospheric concentration". This solves the unit problem in equations (5) and (25).

Thank you for spotting this error. We changed the text on table 2 to read:
"Atmospheric mass of C per mixing ratio            2.123 GtC/ppm "

Equation (8) should then be            $p_{CO2A} = m_A \cdot \varepsilon^{-1}$   (8)

As far as I could see from the code, it is implemented correctly.

Equation corrected as proposed.

In equation (7) the units do not give the desired (Table 1) μ mol/kg. To that end the unit of $M_{\mu mol}$ needs to be changed to gC/μ mol.

Table 2 entry corrected to read:
"mass of DIC per micromole            12.0107 $10^{-6}$ gC/μ mol"

Very confusing is also the use of different time units like in equation (9). Carbon fluxes are measured per year and heat fluxes per second (W=J/s). Nevertheless, the integral limits are in both cases $t_0$ and $t$. I could not find any correction term in the manuscript. In the code, this correction seems to be made.

We prefer to continue to use units of "year" for carbon fluxes and units of "Watt" for heat fluxes. These units are commonly used in the literature. For example, carbon emissions are typically tabulated as annual emissions in GtC/yr, while GHG radiative forcing are given in W/m2.
The following explanation is added after eq. 16:
"Note that for compatibility with commonly used units, carbon fluxes are expressed in Gt per *year*, while heat fluxes are expressed in Joule per *second* (Watt) in equations (15) and (16), respectively."

In equation (10), the unit results in W, not in PetaW as stated in Table 1. Also here a correction term is necessary. Again, this seems not to be an issue in the code.

The unit is now indicated as W. Note that the equation does not depend on the unit used for heat fluxes.

Linear equations and IRFs
Equation (14) is only true if $m(t_0) = 0$. The general equation for the state $m$ at time $t$ is

$$m(t) = r(t - t_0) \, m(t_0) + \int_{t0}^{t} f(t) \, r(t - t') \, dt' \quad (2)$$

Since the authors use equation (14) to compute perturbations with an equilibrium value $m(t_0) = 0$, this does not lead to problems, but the way equation (14) is described is mathematically not correct. I have the feeling this happened, because the authors from the beginning had a perturbation with equilibrium equal to zero in mind, but started the section then with a slightly more general set up. Line 17 on page 5 does not mention perturbations.

We have changed the notation by extending the lower integration limit to negative infinity. This avoids the issues mentioned by the reviewer, and can be applied to perturbations or totals (the latter is necessary to capture the response of terrestrial carbon stocks to warming).

The infinite time scale issue
When equation (20) is inserted in (14) to obtain equation (21), $\alpha_\infty$ somehow disappears.
As soon as $\alpha_\infty \neq 0$ (ocean IRF), a term is missing in equation (21). The equation should then look like

$$m(t) = \int_{t0}^{t} f(t')\alpha_{\infty}dt' + \sum_{k} \int_{t0}^{t} f(t')\alpha_{k} \exp(-(t-t')/\tau_{k})\ dt' \qquad (3)$$

If $f(t')>c>0$, then the first part of the equation goes to infinity as $t \to \infty$ and the perturbation grows indefinitely. If some constant share from a constant input is never going to be decayed, this share accumulates forever. Also the carbon coming from carbon conversion in the ocean model (page 7, line 20), is going to decay at some point. A constant share of remaining carbon should not result from a multiplication with an input flux coming from the atmosphere. The same explosion effect can be seen in equation (A1). Also in Equation (A14) the to $\alpha_{\infty}$ associated term $1/2\ \alpha_{\infty}(\Delta t)^{2}$ is missing. It cannot already be included in the present sum, because $B_{\infty} = 1/2\ \alpha_{\infty}$ is never going to be multiplied with an exponential. I do not know how this problem is handled in the implementation of the model.

- The term for the infinite time scale is now explicitly added in eq. (21) to (24) and in section A1
- The term for the infinite time scale is correctly treated in the code.
- We added the following text on p6, line 22: "We emphasize that the implementation considering only the partitioning of excess carbon between atmosphere, land and ocean (hence $a_{\infty} \neq 0$), neglecting ocean sediment-interactions and weathering flux perturbations, is only valid for time scales shorter than about 2,000 years. "

The theoretical issue

In my opinion, the theoretical foundations of this model are sound as long as the substitute IRFs are time-independent and also independent of other state variables. However, the great power of BernSCM emerges when temperature or CO2 dependencies are explicitly allowed in the IRFs. I am not perfectly sure, if the theory behind the IRF approach is still valid in this case, even though the simulations show reasonable results.

I think it is important to stress the fact, that equation (14) works for time-dependent forcings, but for time-independent processes only. The impulse response function $r$ here depends only on the difference $t - t0$ of the time $t$ at which we are interested in the perturbation $m(t)$ and the time $t0$ at which the input $f(t0)$ came into the system.

The absolute time $t0$ is not used by the impulse response function $r$. Consequently, the underlying black-boxed process which can be modelled by this approach is assumed to be time-independent (has constant coefficients). If now the IRF depends additionally on temperature, the impulse response function needs to "know" the current time $t0$ and becomes $r(t', t-t')$. A different forcing function $f$ leads to a different system state which then results in a different IRF. The system is inherently non-linear, even though it looks linear. From the code (and unfortunately only from there) I could see that the IRFs are provided as a set of coefficients $a_k$ and a set of time scales $\tau_k$. Probably these numbers result from an analysis of complex simulations (e.g. HRBM). If so, the numbers come from a non-linear model and this very IRF is representative only for this very model run. In a non-linear setting, a different model run (initial value, temperature, sensitivities) could theoretically lead to a very different IRF which is then going to be ported to BernSCM. The analysis in section 4 shows that this does not have drastic influence here, probably in part because the external emission forcing was chosen to be the same (SRES A2).

Additional to this possible dilemma, the IRF comes with additional numbers for temperature sensitivity. These numbers are used in each time step to adapt the IRF in dependence of current mean surface air temperature. As mentioned above, $r(t - t0)$ becomes $r(t', t-t')$. In the derivation of equation (A7), which is crucial for the numerical implementation, this leads to a problem. The term

$$R_{i} = 1/\Delta t \int_{t_{i-1}}^{t_{i}} r(x)dx$$

Becomes

$$\tilde{R}_{i} = 1/\Delta t \int_{t_{i-1}}^{t_{i}} \tilde{r}(t-x,x)dx$$

and the integration becomes much more difficult, in particular if both $a_k$ and $\tau_k$ are temperature dependent.

I am not sure whether this problem can be discussed away by numerical means or even by purely theoretical considerations, but at the moment I have a strange gut feeling about this issue.

We agree with the reasoning of the reviewer. The violation of linearity for temperature-sensitive IRF-parameters was pointed out by both reviewers. This issue is possibly related to a badly placed remark on this temperature sensitivity in the context of IRF-integrals on top of page 7. In fact, these parameters can be varied only in the context of a box-model interpretation, and we failed to point this out clearly. The text was modified and extended for clarification:

"The IRF representation is, strictly speaking, only valid if the described subsystem is linear. Then, the response function $r$ does not depend on time and on state variables. In the BernSCM, major nonlinearities in the carbon cycle, namely air-sea gas exchange and the nonlinear carbonate chemistry and changes in NPP in response to changes in environmental conditions are treated by separate nonlinear equations (equations (4) and (5)), while surface-to-deep ocean transport of carbon and heat and respiration of carbon in litter and soils are viewed as approximately linear processes using IRFs. Yet ocean circulation and the respiration of carbon from soil and litter is likely to change under global warming, violating assumption of linearity. In practice, the IRF representation remains a useful approximation as long as the impact of associated nonlinearities on simulated atmospheric CO2 and temperature remain moderate.

The interpretation of the IRF representation as a box model provides a starting point for considering nonlinearities in the response. To account for nonlinearities, the response time scales $\tau_k$ and the coefficients $a_k$ may be gradually adjusted as a function of state variables such as temperature. As the integral form (13) involves integration over the whole history at each time step, changing parameters along the way would result in inconsistencies. In contrast, the differential or box-model form (21) does not depend on previous time steps. Changing the model parameters from one step to the next thus equates to applying a slightly different model at each time step. Within each time step, the parameters remain constant, and the solution for the linear case applies. As time steps are small compared to the whole simulation, this discretization yields accurate results, which is confirmed by the close agreement between the different time resolutions shown in Figure 6 (formerly 4).

Varying coefficients have been successfully implemented and tested for the HRBM land component and its decay IRF (Meyer et al., 1999). In this way, the enhancement of biomass decay by global warming is captured (s.a. Appendix A and section 3.1). In such a modification, the advantage of the IRF and the equivalent box model representation - the faithful representation of the characteristic response time scale of a model system - is largely maintained, while at the same time the impact of time and state-dependent system responses on simulated outcomes is approximated.

**Technical corrections**

The following corrections were all incorporated as suggested. Specific clarifications are provided below for a few points.

• in general: Punctuation around equations is missing very often, in particular full stops or commas after the equations when necessary.

Done. Comma added where appropriate (but not full stops)

• in general: Some abbreviations are never introduced, e.g., HRBM, HILDA. Sometimes the explanation of the abbreviation comes late in the text. This happens in particular when reading the figure captions and figures are referred to in more places. Maybe this is hard to circumvent without destroying the text flow.
• There is a mix of British (analyse) and American ("behavior") English.
• in general: It is difficult to find out which constant means what since tables 1 and 2 are not complete. Some terms are explained in the text, some in the tables.

The tables were revised and completed. Table 2-4 now list all model parameters.

• page 1, line 14: "in an spatially"
• page 2, line 34: "of BernSCM a an IAM component"
• page 2, line 35: "managment"

• page 3, line 1: "cycle assessments(Levasseur et al., 2016)":
space before parenthesis

• page 3, line 26: What does "LULUC" stand for?
• page 4, line 4: "by" or colon missing at the end?
• page 4, line 7, equation (4): How is $\varphi_{NPP}$ defined?

> Function definitions are now given in a new Appendix 1. The purely symbolic equations 4 and 6 were deleted.

• page 4, line 11: "Ao" instead of $A_O$, "eps" is probably meant to be "$\varepsilon$"
• page 4, line 14, equation 8: How are the functions $\psi$ and $\chi$ defined?

> Function definitions are now given in a new Appendix 1. The purely symbolic equations 4 and 6 were deleted.

• page 5, line 1: "and the separation of SAT from radiative equilibrium":
Does "separation" here refer to the difference between $1$ and the ratio $\frac{\Delta T}{\Delta T^{eq}}$? To me the word "separation" is rather confusing in this context.

> Replaced "separation" by "deviation"

• page 5, line 9, equation (12): I could not find a description of $p_{CO_2\,A0}$ anywhere. Typo?
> Typo in line 10 corrected

• page 7, line 5, equation (21): The lower limit of the integral should be $t_0$ instead of $0$.
> All integration limits have been changed to -infinity.

• page 7, line 15: "$a_{Ok}$" is called $a_k$ in Figure 1 (blue box). Also in the red box the constants are called $a_k$. Only in the green box they are called $a_{Lk}$. Similar problems with $\tau$.

> Figure 1 was changed as suggested.

• page 7, line 20: Which model from Table 3 is here referred to? The Bern2.5D or the 4-box Siegenthaler and Joos?

> Reference added on line 21

• page 8, lines 9-10: "here, the IRF substitutes for the HILDA ocean model, and the HRBM land biosphere model are used for the standard setup": It took me a while to understand this phrase. Maybe an additional "for" in front of "the HRBM" and omitting the comma are helpful?
• page 8, lines 13-14: "and the dependency of land C on temperature ($f_{decay}$) increases with warming, eq. (2))": From equation (2) I cannot see what happens with warming. Going to equation (19), I can see that it depends very much on $f_{NPP}$. This is defined in equation (4) and depends heavily on $\varphi_{NPP}$, which is not explained at all.

> - Term "eq. (2)" deleted
> - Functions are now given in the MS

• page 9, line 4, equation 25: An interchange of $\beta_O$ and $\beta_L$ on the right hand side makes it better comparable with the left hand side and the following text.
• page 9, line 7-8: "$\beta$ is the change in carbon stored (in GtC)":
Following Table 3, $\beta$ has the unit GtC/ppm and in Friedlingstein et al. (2006) it is referred to as "sensitivity of land carbon storage to atmospheric $CO_2$". If I understood equation (25) correctly, this is a more precise and less confusing description. The same holds for gamma.
• page 9, line 26: I did not immediately recognise "airborne fraction" as a technical term. Maybe a short explanation could avoid confusing non expert readers.
• page 10, line 1: "(Figure. 3)"

• page 10, line 1-2: Why different units?
• page 10, lines 3-5: Looking at Figure 3 (upper panel), after 100 years all simulated values are greater than 0.3. Where does the value 0.3 from the text come from?

        Value changed from 0.3 to 0.4

• page 10, lines 5-7: "For AF simulated with BernSCM, the standard coupled setup is close to the IRFMIP multimodel median, but the BernSCM uncertainty range is asymmetric. The IRFMIP multi-model range is similarly asymmetric.:" The word "but" confuses me, because the "IRFMIP multi-model range is similarly asymmetric".

        Text changed.

• page 12, line 21: "structural simplicitly"
• page 13, line 15, equation (A4): The integral limits are interchanged. They do not change with the parameter transformation, because $dt_0 = -dx$ makes for a second sign change.

        The equation has been removed in response to reviewer 2.

• page 15, line 2, equation (A14):
Maybe it is better to write $(\Delta t^2)$.

        Equation removed in response to reviewer 2

• page 15, line 13: "explicite"
• page 15, line 20: "Equations (A1,A2)"
Space between A1 and A2?
• page 16, line 13, equation (A20):
Maybe $(m_{S_n} - m_{S_{n-1}})$ is correct? I am not sure.

        The equation in the manuscript is correct. An increase of CO2 in the surface layer reduces C uptake; the term belongs to $-\varepsilon\, p_{S,n-1}^{CO_2}$ (hence negative).

• page 17, line 12: "explicite"
• page 17, line 19: "Equations (11,10, A28)"
Space between equation numbers?
• page 17, line 24, equation (A30): Is it correct that $f_{O_n}$ appears twice in this formula?

        It should be $f_{O_{n-1}}$ in one case. Thank you for spotting this mistake.

• Figure 4: For me it is impossible to differentiate between dashed and dashed-dotted lines here. Maybe a different colour/line-style scheme could help here. Since the differences resulting from the use of different numerical schemes are almost invisible anyway, one could even go without trying to make them visible and simply mention that the differences are small. On the other hand, the point the authors want to emphasize here, is that due to the very small differences, the fastest scheme can be implemented. This leads to the entire appendix and the Fortran implementation. So it is rather an important point.

        Figure 4 (now 6) has been updated as suggested.

• Table 1: "$f_A$: net flux to atmosphere flux"
• Table 1: "$f_{deep}$: Flux mixed layer to deep":
Why does "Flux" start with capital F? Missing "ocean" at the end?
• Table 2: Capitalization of first word in second column inconsistent?
• Table 2: From the units I think $c_p$ should be called "specific heat capacity".
• Code: Why is in the file parLandHRBM.inc the first weight negative? If I understood correctly, those weights are the $a_{L_k}$ values which here nicely sum to one, but how do you distribute a negative share of incoming carbon?

This follows from the original reference (Meyer et al., 1999). The response of several thousand reservoirs (boxes) of the original HRBM model is approximated by five boxes arranged in parallel. The timescales are 0.2, 1.4, 8.9, 74.1 and 253.7 yr. The pool with the short overturning time of 0.2 yr exhibits a very small (-1.3 GtC) negative inventory at equilibrium as noted by the reviewer. This solution is typical for models where material is transferred through successive reservoirs. The two boxes with the smallest time scales of 0.2 yr and 1.4 yr may be combined for decadal to century scale scenario calculations. They yield together a positive inventory of more than 30 GtC.

**Anonymous Referee #2**

This paper by Strassmann and Joos presents the reimplementation of the Bern Simple Climate Model (BernSCM), a reduced form model of the anthropogenic perturbation of the carbon-climate system. This is a historic model for the community, since it and its offspring have been used since the IPCC SAR. This new implementation is useful for the community, especially as this paper focuses on transparency and the model's codeis provided in an open-source format.

Thank you

Being an old model, the BernSCM ignores some relatively recent developments in climate sciences and modeling. In itself, it is not so much of a problem, as the authors leave the door open to further development of the model, both in the manuscript and in the model's code. However, mention and discussion of these caveats is required, especially regarding some specific points I develop below.

Please see our answers below to the specific points raised by the reviewer.

I also believe that the paper could benefit from a more careful rewriting, especially for some sections that I had to read several times – and I am still not 100% sure of what is done in some parts of the paper! In all honesty, some parts give the impression that the authors were in a rush for writing the paper.

We added additional text and references as detailed below to help the reader and the reviewer to better understand the content of the manuscript. We note that reviewer 1 came to a different conclusion and states that "*Apart from some minor exceptions, the manuscript and in particular the appendix and the provided Fortran code of the model are carefully prepared*."

The following text was added in the introduction (p2, l17 of submitted MS); please see also answer to the specific points:
"The BernSCM (Figure 1) is designed to compute decadal-to-millennial scale perturbations in atmospheric $CO_2$, in climate and in fluxes of carbon and heat relative to a reference state, typically preindustrial conditions. The uptake of excess, anthropogenic carbon from the atmosphere is described as a purely physico-chemical process (Prentice et al., 2001). As in pioneering modeling approaches with box-type (Revelle and Suess, 1957;Oeschger et al., 1975) and general ocean circulation models (Sarmiento et al., 1992;Maier-Reimer and Hasselmann, 1987) 
[revised manuscript text omitted]

So I fear publication can only be recommended if the few scientific issues I raise below
are answered/discussed, and if the text itself is improved.

**Major points:**

**1.** My first point concerns the use of the same IRF parameters for the ocean carbon

cycle and the climate system. If I understand it well, the function $ro$ is the same for determining the ocean C sink and the temperature change, e.g. in equations (15) and (16). Although it would seem intuitive to use the same function, because – obviously – we are talking about the (same) world's ocean in both cases, I see several issues in doing so.

First, I am not quite sure one can assume that the diffusion process is the same for heat and for actual material such as carbon. (The assumption seems more reasonable for convection.) But more importantly, the biological pump does not affect heat transport, while it does for carbon. (Although, I am not sure whether there was a biological pump at all in the models used to calibrate the $ro$ function – another thing worth being mentioned.)

> The model does not include a representation of the marine biological cycle as discussed in the answer above.

Second, global patterns of heat uptake vs. carbon uptake are different. This means that one unit of incoming $fo$ is dispatched differently than one of $f_{HO}$, at the scale of the global surface ocean. Therefore, it is likely that each of them is affected differently by the oceanic circulation. For the climate response, it is also known that this pattern affects an internal feedback (the ocean heat uptake feedback) in a way that changes the apparent time-scales of the climate response, see e.g. Geoffroy et al. (2013b) and references therein.

> We modified the text in the discussion (p11, l 31 of submitted MS) to read.
> "Ocean transport is known to vary under climate change with some consequences for heat and carbon uptake (Joos et al., 1999). Here, we applied time-invariant ocean transport parameters ($a_{o,k}$, $\tau_{o,k}$). It is in principle possible to represent temperature dependency of ocean transport in a similar way as it is done for the climate dependency of heterotrophic respiration for the HRBM land biosphere substitute model (Meyer et al., 1999). In the current BernSCM version, the same IRF parameters are applied for the transport of carbon and heat from the surface ocean to the interior ocean. Thereby, it is implicitly assumed that the spatial pattern of change is the same for temperature and carbon. This appears to be a reasonable first-order approximation on decadal-to-century timescales as perturbations in temperature and carbon show similar patterns with decreasing perturbations from the surface to depth. In future efforts, one may differentiate the ocean IRF for heat and carbon, in particular when more information from long-term multi-century to millennial-scale ESM simulations becomes available. The application of the same IRF for carbon and heat in individual model runs implies that modelled carbon and heat transport tend to be physically consistent. In contrast, some other simple models employ different transport parameters for heat and carbon and varied these parameters independently in probabilistic studies."

Third, the typical climate IRF only has two time-scales (e.g. Geoffroy et al., 2013a), and these are quite different from the time-scales from Joos et al. (1996). And, maybe more importantly, the typical two-box climate model implied by the typical climate IRF (Geoffroy et al., 2013a) includes a bidirectional exchange of energy between the surface and deep oceans. This is not the case in the assumed formulation presented here. There are some fundamental reasons for not having this bidirectional exchange for the carbon cycle: the so-called 'ocean invasion' is a slow process, and ultimately there is a sink of C in the deep ocean that involves geological chemical reactions (and time-scales). But can this be also applied to the climate system and heat transport?

> The statement by the reviewer is incorrect. The BernSCM employs bidirectional exchange of carbon and heat. For example, rearranging equation 10 of the submitted MS yields the ocean heat uptake as difference between Radiative Forcing, $RF$, and the response, $\lambda \, \Delta T$, with $\lambda$ the climate sensitivity in W m$^{-2}$ K$^{-1}$:
>
> $$F_o^H = RF - \lambda \cdot \Delta T \quad \text{with} \quad \lambda = RF / \Delta T_{eq}$$
>
> Similarly, the net flux of carbon into the ocean is the results of an uptake flux proportional to the partial pressure (or more correctly the fugacity) of $CO_2$ in the atmosphere and a return flux proportional to the partial pressure of $CO_2$ in the surface ocean (Eq. 5). This feature is now discussed in the discussion (see answer to general comment above).

In addition, the following text is added (p5, line 7 of submitted MS):
"Equation (10) describes ocean heat uptake as difference between *RF* and the climate systems response, $\lambda \, \Delta T$, with $\lambda = RF/ \Delta T_{eq}$ climate sensitivity expressed in W m$^{-2}$ K$^{-1}$.

In the discussion, we added (p 11, l 32):
"The BernSCM model may be extended to model perturbation in the signatures and exchange fluxes of the carbon isotopes $^{13}$C and $^{14}$C as demonstrated in earlier work (Joos et al., 1996). This was not implemented here to keep the code as simple as possible and as most potential users are likely concerned with the evolution of climate and atmospheric $CO_2$."

There is no fundamental reason to not consider the bi-directional flux for carbon. Neglecting the return flux of excess carbon from the surface ocean to the atmosphere corresponds to assuming an infinitely large ocean and infinitely fast mixing between the surface and the deep ocean. Such an assumption leads to erroneous and misleading results.

Further, there is no reason to assume that heat transport by the ocean is governed by two time scales only.

We added the following text in the discussion p11, l 32:
"A distribution of time scales applies to ocean transport processes as evidenced by observations of transient and time dependent tracers such as chlorofluorocarbons, bomb-produced and natural radiocarbon and biogeochemical tracers (Olsen et al., 2016;Key et al., 2004). This continuum is sometimes approximated by one time scale, also termed heat uptake efficiency (e.g. (Gregory et al., 2009)) by two time scales, as in (Geoffroy et al., 2012b). The one to two time scale approximations were used to analyze relatively short Earth System Model simulations that do not yet reveal the multi-century response time scales of the deep ocean. We note that the equivalent ocean depth of the simple energy balance model of (Geoffroy et al., 2012b) for their AOGCM ensemble is only 1182 m compared to a mean ocean depth of about 3800 m. The ocean IRFs used in the BernSCM are derived from long simulations with ocean-only or simplified models. The range of distinct time scales used to construct the IRF faithfully approximates the sub-annual to multi-century response continuum of the parent models as shown in earlier work (Joos et al., 1996). Further, the BernSCM IRF model represents the heat capacity of the entire ocean."

Therefore, I believe this is an assumption made by the authors that $ro$ can be applied to the climate system as well. Despite all of the above, it may still be acceptable. But it should be presented as such, and it also warrants a discussion in the text. Additionally, the response to a step of radiative forcing (typically 4x CO2) of this climate model has to be compared to that of more complex models. I strongly suggest adding a (sub)figure in which the BernSCM climate response is compared to that of CMIP5 models, taken e.g. from Geoffroy et al. (2013b). This would complement figure 3.

The new Figure 3 demonstrates that the BernSCM temperature response falls well within the response of more complex models.
Thank you for suggesting this additional figure. We carried out additional simulations where $CO_2$ is prescribed to increase either exponentially (linear increase in RF) or abruptly to reach 4xCO2 (Figure 3). We compare the outcome in terms of realized warming fraction with the compilation by Frölicher and Paynter (2015) for EMICs and CMIP5 AOGCMs (please refer to newly added section 3.2).

**2.** I was very troubled by section 3 and how the carbon-climate feedbacks are represented/ investigated with BernSCM, in relation with C4MIP. At first, I thought BernSCM was trying to emulate the C4MIP models' sensitivities (which would have been a new feature).

In the end, my understanding is that the uncertainty range provided e.g. in table 4 is obtained by combining variations of: (i) the ocean model, 2 options; (ii) the land model, 2 options; (iii) the experimental setup, i.e. coupled/uncoupled/Tonly/Conly, 4 options. That is a total of 2× 2× 4=16 configurations. But my concern, here, is that I think that

turning a process on or off can hardly be considered a new configuration of the model. Therefore, although the results shown e.g. in figure 3 or 4 are interesting, the ranges provided in table 4 are artificial and misleading.

More generally speaking, the text should make it clear that there are not many parameterizations available for the model, and so it does not cover the full range of existing multi-model uncertainty (and therefore, it cannot be used in a probabilistic fashion). Again, it is not so much of a problem in itself, but this has to be made very clear.

> Done. We incorporated section 3 into section 4 "Illustrative simulations with the BernSCM" and reorganized section 4 to avoid a potential misunderstanding. This is done by adding the following subsection headings in section 4:
> 4 Illustrative simulations with the BernSCM
> 4.1 Model setup for sensitivity analyses and uncertainty assessment
> 4.2 Fraction of realized warming and idealized forcing experiments
> 4.2 Impulse response experiment
> 4.3 Carbon cycle-climate feedbacks
> In section 4.1 we included the existing text from p 8 line 5 to 29 of section 3 followed by the following text:
> "We performed simulations with these different setups.  In section 4.2, we probe the time scales of the temperature response in simulations where atmospheric $CO_2$ is abruptly (instantaneously) quadrupled or by increasing CO2 radiative forcing linearly within 140 years. In section 4.3, we probe the response of the coupled system to a pulse-like release of 100 GtC into the atmosphere. Finally in section 4.4, we analyze carbon cycle-climate feedbacks relying on simulations over the industrial period and for the SRES A2 scenario. BernSCM results are compared with the results from three multi-model intercomparison projects: the Climate Model Intercomparison Project 5 (CMIP5) with results as summarized by (Frölicher and Paynter, 2015); an analysis of carbon dioxide and climate impulse response functions …"

> Section 3.2 describes the results for the 4xCO2 simulations requested by reviewer 2 (please see the revised manuscript (attached).

> Section 4.3 on IRF experiments includes the text from page 9, line 25 to p10, l17 of the originally submitted MS

> Section 4.4 on feedback analyes includes the text from p8, l30 to p9, l16 followed by the paragraph on p10, l18 to p10, l22. Then the paragraph on p9, l17 to l19 is added before continuing with the text from p10, l23 to p11, l2.

> The caveat that only a limited set of model versions is available is already explicitly dicussed in section 5, p11, line 27 or original MS ("Currently, a limited set of substitute models is available …")

> We somewhat disagree with the reviewer on the potential use in probabilistic assessment. As with any model, the parameters of the BernSCM (including those of the land and ocean IRF) can be varied using Latin Hypercube sampling or similar and simulation results weighted with observational constraints as for example demonstrated by Steinacher et al. (2013). We note that spatially-explicit and dynamic Earth System Models of Intermediate Complexity offer a much greater potential in probabilistic assessment than the current crop of simple models.

**3.** I have some trouble with the way the solving of the differential system is presented, but more importantly I believe there is a mistake with how the temperature-dependent parameters are implemented.

I am not convinced by the lengthy demonstration of appendix A1. Equations (A2) and (A3) are the 'results' of this section, and I believe the following demonstration is not needed. Equations (A2) and (A3) can simply be obtained by using the 'exponential integrator' method to solving a first-order differential system. Although not everyone may know this method, it could be summed up in one or two equations (and appropriate references) rather than be re-demonstrated from scratch.
Equations (A2) and (A3) are simply obtained by 'reminding the reader' that the solution to the differential system:

$$\frac{\mathrm{d}m}{\mathrm{d}t} = -\frac{m(t)}{\tau} + aF(t) \qquad (1)$$

can be discretized by multiplying by $\exp(\delta t/\tau)$ and integrating between $t_n$ and $t_{n+1} = t_n + \delta t$:

$$m_{n+1} = \exp-\frac{\delta t}{\tau}m_n + \int_0^{\delta t} \exp(-\frac{\delta t - s}{\tau})aF(t_n + s)\mathrm{d}s \qquad \hookleftarrow_2\hookleftarrow$$

where $m_{n+1} = m(t_{n+1}) = m(t_n + \delta t)$, and $\delta t$ is the time step.

The above equation is exact, but can hardly be solved. It is usual to assume that $F$ is constant over the small time period of $\delta t$, which leads to the solution:

$$m_{n+1} = \exp(-\frac{\delta t}{\tau})m_n + \tau(1 - \exp(-\frac{\delta t}{\tau}))aF(t) \qquad \hookleftarrow_3\hookleftarrow$$

which is basically equation (A2) and (A3) combined. $\tilde{\ }t$ remains to be chosen, e.g. to be $t_n$ (forward method), $t_{n+1}$ (backward), or any other fancier method possible. When assuming $\delta t$= 10 yr and $a\ F(t)$ is linear between $t_n$ and $t_{n+1}$, one immediately finds the $\delta t^2$ equations.

So far, no fundamental problem with the authors' equations and text. I just believe it could be written in a more efficient and straightforward way. But a problem arises when one assumes that the time-scale $\tau$ varies with time (through e.g. temperature) so that we have in fact $\tau = \tau_0 + \Delta\tau\ (t)$. The exponential integrator method can still be applied, albeit by using $\tau_0$ and not $\tau$ in the exponential function.
To do so, it is easier to rewrite the differential equation as:

$$\frac{\mathrm{d}m}{\mathrm{d}t} = -\frac{m(t)}{\tau_0 + \Delta\tau} + aF(t) \qquad (4)$$

$$= -\frac{m(t)}{\tau_0} + -\frac{m(t)\Delta\tau}{\tau_0(\tau_0 + \Delta\tau)} + aF(t) \qquad (5)$$

which completely changes the exponential integrator form:

$$m_{n+1} = \exp-\frac{\delta t}{\tau_0}m_n + \int_0^{\delta t}\exp(-\frac{\delta t - s}{\tau_0})aF(t_n + s)\mathrm{d}s$$
$$+ \int_0^{\delta t}\exp(-\frac{\delta t - s}{\tau_0})\frac{m(t_n + s)\Delta\tau(t_n + s)}{\tau_0(\tau_0 + \Delta\tau(t_n + s))}aF(t_n + s)\mathrm{d}s \qquad \hookleftarrow_6\hookleftarrow$$

leading to:

$$m_{n+1} = \exp(-\frac{\delta t}{\tau_0})m_n + \tau(1 - \exp(-\frac{\delta t}{\tau_0}))aF(t') + (1 - \exp(-\frac{\delta t}{\tau_0}))m(t')\frac{\Delta\tau(t')}{\tau_0 + \Delta\tau(t')} \hookleftarrow_7\hookleftarrow$$

The latter equation raises the issue that it is virtually impossible to use with a backward approach since $\Delta\tau\ (t_{n+1})$ is not known. But a bigger issue is that, if I understand it correctly, the authors do not use this equation nor any equivalent. I believe they simply apply the equation of the case with constant $\frown$ but with a value of $\frown$ that changes through time. That is, they use the following equation:

$$m_{n+1} = \exp(-\frac{\delta t}{\tau_0 + \Delta\tau(t)})m_n + (\tau_0 + \Delta\tau(t))(1 - \exp(-\frac{\delta t}{\tau_0 + \Delta\tau(t)}))aF(t') \quad (8)$$

instead of the one above.
Unless the authors can prove the difference between the two is negligible, I am afraid
there is a fundamental mistake in the solving of the model.

> We agree that appendix A1 was unnecessary lengthy. The section was rewritten in a simpler way according
> to the reviewer's suggestions. Thank you.

> The violation of linearity for temperature-sensitive IRF-parameters was pointed out by both reviewers. This
> issue is possibly related to a badly placed remark on this temperature sensitivity in the context of IRF-
> integrals on top of page 7. In fact, these parameters can be varied only in the context of a box-model
> interpretation, and we failed to point this out clearly. The following text was added to clarify this point
> (please see the response to reviewer 1 for the full text changes):

> "The interpretation of the IRF representation as a box model provides a starting point for considering
> nonlinearities in the response. To account for nonlinearities, the response time scales $\tau_k$ and the coefficients
> $a_k$ may be gradually adjusted as a function of state variables such as temperature. As the integral form (13)
> involves integration over the whole history at each time step, changing parameters along the way would
> result in inconsistencies. In contrast, the differential or box-model form (21) does not depend on previous
> time steps. Changing the model parameters from one step to the next thus equates to applying a slightly
> different model at each time step. Within each time step, the parameters remain constant, and the solution
> for the linear case applies. As time steps are small compared to the whole simulation, this discretization
> yields accurate results, which is confirmed by the close agreement between the different time resolutions
> shown in Figure 6 (formerly 4)."

**4.** I believe the model should be completely described in the paper. I mean: formulations
for e.g. functions $p_{CO_2 S}$, $\phi$, $\chi$, as well as all the parameter values should be
given. The model is relatively simple, and there are not that many parameters. Even if
the values can be accessed in the code, the fact that this paper is a model description
makes it necessary to be as exhaustive as possible.

> Done. Parameters for the formulations are now given in the appendix and corresponding tables.

**Minor points:**
p. 1 (sec. 1): SCMs have many more usages than what is given here. Generally
speaking, I find that the citations of this paper are too self-centered. I think everyone
acknowledges the importance of the original Joos et al. (1996) paper, but much has
been done since then regarding IRFs.

> Done. We have complemented the list of potential applications on p. 2 l.2 and provide references to other
> simple box-type and IRF models: "Another application of simple models (e.g., (Enting, 1990;Enting et al.,
> 1994;Oeschger et al., 1975;Siegenthaler and Oeschger, 1984;Huntingford et al., 2010;Smith et al.,
> 2017;Tanaka et al., 2007;Bruckner et al., 2003;Joos and Bruno, 1996;Hooss et al., 2001;Urban and Keller,
> 2010;Good et al., 2011;Wigley and Raper, 1992;Raupach et al., 2013;Boucher and Reddy, 2008) is to
> compare, analyze or emulate more complex models ((Meinshausen et al., 2011;Raper et al.,
> 2001;Geoffroy et al., 2012a;Thompson and Randerson, 1999;Geoffroy et al., 2012b). Simple models also
> play a significant role in previous assessments of the Intergovernmental Panel on Climate Change (e.g.,
> (Harvey et al., 1997))."

p. 3, l.13: The "essentially linear behavior" is an assumption of the model.

> Done. Text modified to read: "but in first order linear behavior"

p.3, l. 16: IRFs are indeed equivalent to box-models, albeit with constant parameters!

    Thank you for your confirmation

p.3, l.25: The non-inclusion of LULCC could be discussed a little.

    Done. The following text is added: "Human impacts on the land biosphere exchange including land use and land use changes are not simulated in the present version, and treated as exogenous emissions ($e$). These emissions may be prescribed based on results from spatially-explicit terrestrial models."

p.6 l.10: It is probably better to give all the equations, even if very similar.

    Done. Text modified to read: "Similarly, equation (16) closes the heat budget equation (9) for the surface ocean."

p.6 l.20: "conversation" => "conservation" (probably many typos I missed...)

    Done. Typo corrected

p.6 l.22: I don't think it is $10_4$ or $10_5$ kyr. Unit is probably yr.

    Done. Unit correct to yr

p.7 l.2: At this stage, it is very unclear whether the response based on HRBM is a usual linear IRF calibrated with climate-carbon feedbacks on, so that those are linearized within the IRF, or if the time-scales of the response are indeed interactively changed by temperature during the simulation. Note also that I don't think the name "IRF" can be given to a model with time-varying parameters. I believe an IRF is the integrated form of the differential equation, which can be obtained only when the parameters do not vary with time. When they do, there is no integrated form, and the model is just a box model.

    Done. Text modified to read: "In contrast, the parameters of the IRF-derived box model representation of the HRBM land biosphere are interactively modified during the simulation by a temperature dependent factor. In this way, the enhancement of biomass decay by global warming is captured (s.a. Table 3 and section 3)."

p.7 l. 13: Similarly, I would question the fact that the equation shows that IRF and box model are equivalent. I think they are per definition. The only difference being that one is the integrated form of the other.

    Done. Text changed to read: "Equation (22) represents the IRF by a box model, ..."

p.7 l. 20: Can cite Li et al. (2009) who provide a nice discussion on the (over)interpretation of those parameters.

    Done. Reference to (Li et al., 2009)added.

p.7 l.25: It is more than 'they can be viewed'. Per construction, IRFs show the exponential eigenmodes of the system they are calibrated upon. Raupach (2014) or Enting (2007) provide some insights on this.

    Done. Text modified to read: ".. are equivalent to .. and may also be interpreted in the context of the Laplace transformation (Raupach et al., 2013;Enting, 2007)".

p.8 (sec. 3): I really find this section difficult to apprehend. It would benefit from some re-organizing, e.g. with a subsection on the beta/gamma framework, and then one on what it gives when applied to BernSCM. This is also the part that made me wonder whether C4MIP models were emulated or simply used for comparison.

    Done. Please see answer to major comment 2.

p. 8, l.16: Table 3 does not provide any parameter value

      All functional forms and parameters are now given in the new Appendix A and corresponding tables.

p.8, l. 28-29: Please, name those simulations "T-only" and "C-only". The dash makes
a lot of difference when reading the text that follows!

      Done.

p.9, l.9: I don't think alpha is the "transient climate sensitivity" in the usual sense. Find
another name.

      Done. Text modified to read "linear transient climate sensitivity to CO2 ($^{\circ}$C per ppm)" as in Friedlingstein et
      al. (2006).

p.9, l.10: Which original paper?

      Done. Reference to Friedlingstein et al. (2006) added.

p.9, l.31: The "combinations" remain quite unclear.

      The combinations are given in the caption of Figure 4 (formerly 3).

p.10, l.1: Inconsistent temperature units (this is in the whole paper).

      Done. Notation adjusted.

p.10, l.2: More important comment related to my first major points. The choice of a
climate sensitivity does not affect the time-scales of the climate response. However,
it is known that a higher climate sensitivity implies a slower climate system (e.g. Baker
Roe, 2011).

      Done. Please see the new figure to gauge the time scales of the climate system response in the model

p.10, l.9-10: The last bit of this sentence is very uninformative.

      Done. Text deleted.

p.10, l.12: 3.2K

      Done. Unit added.

p.10, l.15-17: I believe the fundamental reasons exposed in my major point number 1
also explain a lot, here. Hence the need to compare the climate response alone, and
not coupled to the carbon cycle as in figure 3.

      A new figure that compares the climate response in isolation is added (new Figure 3).

p.10, l.23: Those sensitivities are not defined

      Done. Sentence deleted in the revision process

p.10, l.29-32: I don't see the point of those sentences. Yes, the obtained sensitivities
are zero. But this is per construction, since the uncoupled cases are used to investigate
the sensitivity. This relates to my major point 2.

      We prefer to keep this information in the text. Please see above for the response to major point 2

p.11, l.4: I believe it is 0.5K, according to figure 4. Also these values are for a fixed
climate sensitivity. So I wonder how informative they are.

      Done. Text on page 11 line 3 to 5 deleted

p.11 (sec. 5): I don't find all the discussion about BernSCM/C4MIP very convincing, for the reason already exposed above.

> Done. We shortened the discussion and deleted the text from line18 to 23.

p.12, l.1: Yes, but that requires building EOFs on more complex models. Mention and citations needed here.

> Text modified to read: "A potential future application .."

p.12, l.3: Note that regarding precipitation (and likely cloud cover as well), we now know that the response is forcing dependent (e.g. Shine et al., 2015; and referencestherein).

> Done. We added the following text "Patterns of change are generally similar across models for temperature, whereas patterns in precipitation are more uncertain and show greater variability between models (Knutti and Sedlacek, 2013) and are forcing dependent (Shine et al., 2015). We also note that natural variability strongly influences the space-time evolution of climate change (Deser et al., 2012). Patterns may be scaled with changes in global mean surface air temperature as indicated in Figure 1 or dependencies on radiative forcing may be considered (Shine et al., 2015)."

p.12, l.10: Yes. But simple models usable in a probabilistic fashion already exist out there.

> Done. Text added ", although more sophisticated models are available for observation-constrained probabilistic quantification of climate targets (Steinacher et al., 2013;Holden et al., 2010;Steinacher and Joos, 2016)"

p.12, l.23: GWPs and other metrics require inclusion of non-CO2 species. So I'm not sure the sentence here is relevant.

> Text clarified to read: " .. by applying emissions- or concentration-driven simulations."

p. 12, l.26: I don't like the use of "fixed", here. It is e.g. not influenced by external factors such as climate change.

> Done. "fixed" replaced by "ocean transport not influenced by climate change."

p. 13 (sec. A1): As I wrote in my major points 3, I believe this section could be more straightforward.

> We agree. The section was rewritten in a simpler way according to the reviewer's suggestions. Thank you.

p. 13 (sec. A2): This section is awfully complicated! It makes me wonder about several things, and I could not find the answer. Couldn't a solver be used for the backward method? Is the backward method solved with an exact solution, or is the method proposed an approximation? Does it have to be that complicated?
Also, I find the equations extremely difficult to follow. There are four (!!) levels of notation: $U, V, W$ refer to $p_{fk}$ which refer to $A_k$ which refer to the original parameters $\frown_k$ and $a_k$. I am convinced this part could be written (and implemented in the code?) in a much simpler way

> We agree that a solver would be simpler and the equations are complicated. However, we decided for this solution to make the model self-contained and more portable, and we included the equations as such for completeness. The equation systems were solved with a symbolic mathematics software to minimize the risk of mistakes. The number of notation levels has been reduced by the simplification of section A2 (formerly A1).

p.12 (sec. A3): Again, not completely clear how the climate-carbon feedback is implemented. See major point 3.

The section now refers to the formulas for the variable HRBM parameters listed in the new appendix A1.

p.25 (fig. 3): A representation of the land and ocean fractions could be provided. Also, see major point 1: the climate response alone should be shown somewhere (be it within figure 3 or separately).

Done. Two new figures were added, one showing the climate response for an idealized forcing experiment (Figure 3), and one showing the land, ocean, and airborne fractions for the IRFMIP pulse experiment (Figure 5).

p.26 (fig. 4): Maybe show ranges from C4MIP?

Done as suggested.

p.29 (tab. 3): I don't find this table very informative. Parameter values and functional forms should be provided instead.

Done. A new appendix A was added containing all model parameters and functional forms.

p.30 (tab. 4): Using the words "parameters" is one of the things that made me wonder whether C4MIP models were used as input to BernSCM or just to compare outputs. I would call that e.g. "metrics".

Done. Word changed as suggested.

[revised manuscript text omitted]
^{CO_2}_{S,n-1}) + k_g A_O\, \varepsilon\, \left.\frac{dp^{CO_2}_S}{dm_S}\right|_{n-1} (m_{S\,n-1} - m_{S\,n}) \tag{B8}$$

where equations (5,6) were used. Similarly, NPP  as a function of atmospheric carbon is linearized,

$$\Delta f_{NPP\,n} \simeq \left.\frac{df_{NPP}}{dm_A}\right|_{n-1} (m_{A\,n} - m_{A\,n-1}) \tag{B9}$$

using equation (6).

The system is completed with the discretized budget equation (1)

$$m_{A\,n} = m_{A\,n-1} + (e_{n-\frac{1}{2}} - f_{O\,n})\Delta t - (m_{L\,n} - m_{L\,n-1}) \tag{B10}$$

Here, $e_{n-\frac{1}{2}}$ is assumed to be known (though this only applies to the "forward" solution for atmospheric $CO_2$ from emissions, solving for emissions from $CO_2$ is also implemented in the model code).

After calculating the "committed" values $m^{c\Delta}_{L\,n}$, $m^{c0}_{S\,n}$ from the model state at $t_{n-1}$, equations (B7) through (B10) are solved

$$\Delta f_{NPP} = \frac{\left.\frac{df_{NPP}}{dm_A}\right|_{n-1}}{UV + W}\left( m_{L\,n-1} - m^{c\Delta}_L + \Delta t\, e_{n-\frac{1}{2}} + \Delta t\, k_g A_O \left( \varepsilon\, p^{CO_2}_{S,n-1} - m_{A\,n-1} \right.\right.$$
$$\left.\left. + \varepsilon\, \left.\frac{dp^{CO_2}_S}{dm_S}\right|_{n-1} \left[ m^{c0}_S - m_{S\,n-1} + \sum_k p_{fk_O}\left( \frac{m_{L\,n-1} - m^{c\Delta}_L}{\Delta t} + e_{n-\frac{1}{2}} \right) \right] \right) \right) \tag{B11}$$

with the auxiliary variables

$$U = k_g A_O\, \varepsilon\, \left.\frac{dp^{CO_2}_S}{dm_S}\right|_{n-1} \sum_k p_{fk_O} + 1 \tag{B12}$$

$$V = \left.\frac{df_{NPP}}{dm_A}\right|_{n-1} \sum_k p_{fk_L} + 1 \tag{B13}$$

$$W = \Delta t\, k_g A_O \tag{B14}$$

and, after inserting into equation (B6),

$$f_{O\,n} = \frac{k_g A_O}{U + W}\left( m_{A\,n-1} - \varepsilon\, p^{CO_2}_{S,n-1} - \varepsilon\, \left.\frac{dp^{CO_2}_S}{dm_S}\right|_{n-1} (m^{c0}_S - 
[revised manuscript text omitted]

---

## Author Response (AR2)

**Response to the topical Editor's Comments**

Dear Carlos,
Thank you very much for this positive assessment of the revised manuscript! We agree with your opinion and have addressed the issues raised as detailed below.

Topical Editor Decision: Publish subject to technical corrections (19 Mar 2018) by Carlos Sierra
Comments to the Author:
Dear authors,
thanks for preparing a revised version of your manuscript addressing reviewers' comments. The new version is much more improved and addresses well the main issues raised by the two reviewers. I only have a few minor comments that may help to improve the manuscript. Once they are addressed, we can proceed with manuscript for publication.

I also shared the concern of the reviewers about the potential violation of assumptions of the impulse response approach for the time-varying case. This issue is now clarified both in the answers to reviewers' comments and in the main text. However, I think there's still a few minor issues that could help to further clarify the issue.

- At the end of section 2.3, you added new text clarifying that the IRF approach is only valid if the subsystem is linear. Although the emphasis in linearity is very important, I think it is equally important to emphasize that the approach also relies on the assumption of time-invariance of the time-scales of the system. You may want to emphasize this assumption separately from the assumption of linearity since you may still have a linear model with first-order rates that doesn't meet the assumption of time-invariance.

> We have edited the 5th paragraph on page 9 in section 2.3 as follows: "The IRF representation is, strictly speaking, only valid if the described subsystem is linear and the time scales of the system are time-invariant."

- In several parts in the manuscript you make reference to 'box models', and I got the impression that the term may have different connotations in different contexts. Box model may mean very different things to different people. I would prefer if you use a less ambiguous term. For instance, it would be better to say that the IRF can be interpreted as a system of uncoupled first-order ordinary differential equations. Alternatively, you may want to provide a definition of box model when you first use the term.

> We have changed the introduction of the box model in the second-to-last paragraph of the introductory text of section two on page 5:
> "More illustratively, the ocean and land models can be considered to consist of systems of uncoupled first-order ordinary differential equations or "box models", which are an equivalent representation of the IRF model components (Figure 1)."
> and in the second paragraph of page 9,
> "The differential equation system (21) can be considered to consist of several boxes, whereby each box $m_k$ receives a fraction $a_k$ of the input f, and has a characteristic turnover time $\tau_k$ (Figure1). In the following, this is referred to as a ``box model''."

- Eq. 1. In the text before the equation, specify that the budget equation is for 'atmospheric carbon'.

Done

- Eq. 12. In the text before the equation, specify that the equation applies for a linear system with time-invariant rates with time-dependent forcing.

Done

- Appendix A2. In the equation for NPP, is there a negative sign missing in the first exponential term? Please check.

The term is actually positive. However, we noticed that the parametrization ranges should be mentioned and added the following sentences
- For HRBM NPP: "This expression holds up to a $CO_2$ concentration of 1274 ppm and is capped at that value." and "This expression holds up to a SAT increase of 5°C."
- For HRBM IRF: "The temperature sensitivies of the HRBM IRF are parametrized for a warming of up to 5°C."